# Error prediction determines the coordinate system used for the representation of novel dynamics

**Raz Leib[1], David Franklin[1,2,3]\***

[1]Neuromuscular Diagnostics, TUM School of Medicine and Health, Department Health and Sport Sciences, Technical University of Munich, Munich, Germany; [2]Munich Institute for Robotics and Machine Intelligence (MIRMI), Technical University of Munich, München, Germany; [3]Munich Data Science Institute (MDSI), Technical University of Munich, Munich, Germany

**Abstract** Skillful object manipulation requires a representation of the object's dynamics. Despite extensive research, previous studies have not been able to provide a consistent view of this representation in the motor system, with each study providing evidence favoring an extrinsic coordinate system, an intrinsic coordinate system, an object-based representation, or mixtures of these coordinate systems. In a series of experiments, we show that the motor system combines different representations based on their reliability. Specifically, since noise creates an error between planned and executed force production which depends on the arm state, the motor system will rely more on the representation for which the plan is less affected. In addition, we show that the same mechanism predicts the different results made about dynamics representation and thus explains the discrepancies between influential past studies. Overall, we are able to reconcile all of the apparently disparate findings under a single cohesive model of dynamics representation.

**\*For correspondence:**
david.franklin@tum.de

**Competing interest:** The authors declare that no competing interests exist.

## Editor's evaluation

This study provides a valuable new perspective on how motor learning occurring in one state generalizes to new states (for example, a different limb posture). The paper proposes a new model in which different potential coordinate systems for generalization are combined based on their relative reliability. The authors provide convincing evidence for this model, showing that it improves significantly on previous theories in its ability to predict patterns of generalization of motor learning in human subjects.

## Introduction

During object manipulation, such as swinging a jump rope, we experience external forces. To manipulate the object, we need to estimate the mapping between the motion and the resulting forces, and then send appropriate commands to the muscles that compensate for the arm and object dynamics (*Leib et al., 2024*). While this seems trivial, it is still unknown how we represent the dynamic mapping of such tasks, as there are at least three different reference frames that can solve this problem (*Figure 1A*). First, we could use a Cartesian coordinate frame in which the forces are represented as vectors in a coordinate system located externally to the body. Second, we could represent the forces using the shoulder, elbow, and wrist joint torques (*Shadmehr and Mussa-Ivaldi, 1994*). Third, we could represent the forces as belonging to the hand or a hand-held object using the orientation vector connecting the shoulder and the object or hand in space (*Berniker et al., 2014*).

**Figure 1.** The Re-Dyn mechanism for representing dynamics. (**A**) We considered three main types of coordinate systems for dynamics representation namely Cartesian (red), joint (turquoise), and object (yellow) based representations. Each coordinate system depends on the estimation of different state variables, which include joint angular values or hand coordinates, and those estimations can be distorted by neural noise (ε). (**B**) To test which coordinate system is used for dynamics representation we examined how learned dynamics are generalized in space. Participants moved between points in one position in space (gray circles) while experiencing external forces (gray arrows) and we examined the pattern of these forces in other locations in space using a force channel method. The movement in the new location (black circles) required participants to change the posture of their arms. (**C**) For movements in new locations (black circles), each coordinate system predicts a different pattern of generalized forces (black arrows pattern). These predictions are the desired generalized force pattern according to each reference frame and can be different due to the nature of each reference frame. However, since the current state of the arm has some uncertainties due to neural noise, ε, the spatial end-point position or arm orientation is incorrectly estimated (marked using gray arrows). As a result, the motor system might generate different pattern of forces compared to the pattern desired pattern. That is, noisy estimations for end point position (red), joints angles values (turquoise), and hand orientation value (yellow) in the new locations can alter the force pattern for the Cartesian, joint, and object-based representations, respectively. (**D**) Depending on the spatial location of the training and generalization movements, similar noise characteristics can have different effect sizes on the difference between desired and generated force patterns. This can be quantified by calculating the difference multiple times while randomizing the noise value. Based on the force difference distribution for each representation we can estimate the relative contribution of each representation to the generalized forces. This relative weight can be calculated using, for example, the inverse of the distribution variance. (**E**) An example for repeating the calculations for the Re-Dyn predicted weights across multiple directions. For each direction and according to each of the coordinate systems, we simulated distorted force patterns (gray curves) and compared them to the desired force profile marked using a colored curve (red-Cartesian, turquoise- joints, yellow- object). The distribution of the gray curves around the desired curve gives a sense of the distribution of error for each movement direction. (**F**) Based on the example in (**E**), we calculated the force difference variance of each coordinate system for each direction. To calculate this variance, we repeated the procedure in (**D**); that is, for each simulation (gray curve), we calculated the mean force difference from the desired force profile (colored curve). These differences were used to construct the mean force difference distribution and to calculate how variable this distribution is. (**G**) Using the variance profiles in (**F**) we calculated the weights using an inverse-variance formula (see Methods for more details).

One way to distinguish between these representations is by examining how dynamics are generalized in space. In dynamic generalization, participants learn to move under force perturbations *Shadmehr and Mussa-Ivaldi, 1994* followed by testing how the learned dynamics are extrapolated to new locations in space (*Figure 1B*). Since the extrapolated force pattern can differ when using different reference frames, mainly due to the dependency on arm posture in space or lack of it, we can understand which reference frame was used by the brain. Although many studies have used dynamic generalization to address the topic of representation, the results do not provide a consistent view of the coordinate frame used by the motor system. Some studies support a Cartesian-based representation (*Criscimagna-Hemminger et al., 2003*), while others support a joint-based (*Shadmehr and Mussa-Ivaldi, 1994*) or object-based representation (*Ahmed et al., 2008*; *Yeo et al., 2015*; *Franklin et al., 2016*). In addition, some studies suggested that the system uses a mixture of the aforementioned coordinate frames (*Parmar et al., 2011*; *Berniker et al., 2014*). The lack of consistency between these results raises the question of whether the motor system alternate between different representations and if so, what triggers preferring one representation over the other.

We hypothesize that this inconsistency in results can be explained using a framework in which the motor system assigns different weights to each solution and calculates a weighted sum of these solutions. Usually, to support such a framework, previous studies found the weights by fitting the weighed sum solution to behavioral data (*Berniker et al., 2014*). While we treat the problem similarly, we propose the Reliable Dynamics Representation (Re-Dyn) mechanism that determines the weights instead of fitting them. According to our framework, the weights are calculated by considering the reliability of each representation during dynamic generalization. That is, the motor system prefers certain representations if the execution of forces based on this representation is more robust to distortion arising from neural noise. In this process, the motor system estimates the difference between the desired generalized forces and generated generalized forces while taking into consideration noise added to the state variables that equivalently define the forces. For example, for the Cartesian-based representation, the generalized forces depend on the end-point initial and target points for the movement. When the estimations for these points are noisy, the calculated generalized force is different from the desired forces (*Figure 1C*, upper panel). The difference between these noisy generalized

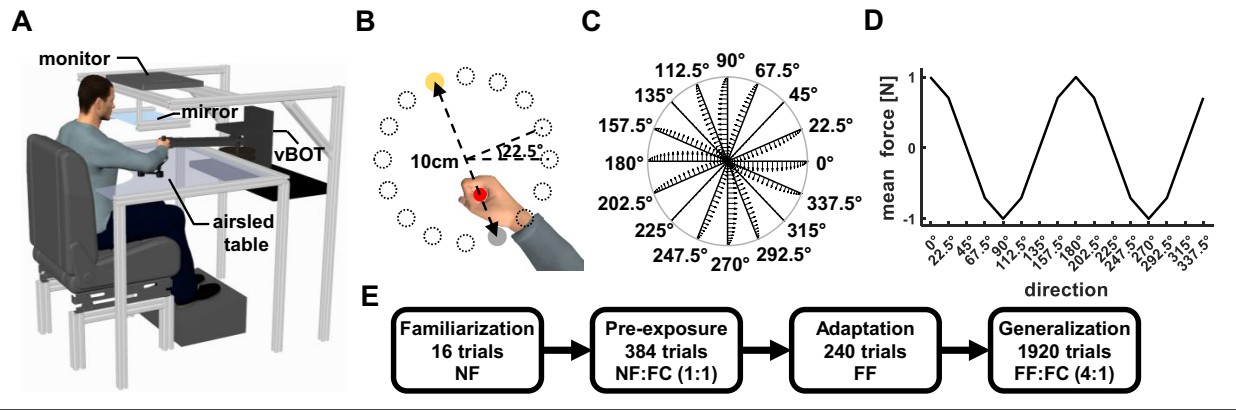

**Figure 2.** Experimental Design. (**A**) Participants sat in front of a robotic manipulandum system (vBOT) while grasping the handle of the robot with their right hand. The participants' arm was supported by an airsled system that reduced friction during movement. The virtual environment was projected on a mirror from a monitor mounted above the movement space. (**B**) Display of the virtual workspace. The participants' hand was represented using a red cursor that was aligned with the hand position. The task was to move from the start point (gray circle) to the target point (yellow circle). The start and target points were always located on the diameter of a 10 cm diameter circle. There were 16 possible starting points spaced evenly on the circumference of the circle. On each trial, only one set of start and target points were presented. (**C**) Endpoint force profiles as a function of movement direction. The forces were calculated using a scaled curl force field. The force profile that appeared in each direction is illustrated from the center to the boundary of the circle. (**D**) Cosine representation of the mean forces of the scaled force field as a function of movement direction. Clockwise forces were set as positive and counterclockwise forces were set as negative. (**E**) Experimental protocol. Initial familiarization phase included 16 trials in which no force was applied to the hand (null field, NF condition). The pre-exposure phase included unconstrained movements (NF condition) and movements in a force channel (FC condition), with a ratio of 1:1 in all three workspaces. In the adaptation phase, participants performed the movements under the force field perturbation (FF condition) only in the training workspace. In the last phase, we tested generalization using force channel trials (FC) that were introduced randomly at the training and test workspaces while participants kept moving under force field perturbations (FF) only in the training workspace. The ratio between force field and force channel trials was 4:1.

forces and the desired forces sets how much the motor system can rely on this coordinate system. In a similar way, such assessment is made for the joint-based representation in which noise is added to the joints angle values and for the object-based representation in which noise is added to the hand orientation (*Figure 1C*, middle and bottom panels). When repeating this process of randomizing the added noise values, while using similar noise distributions across the different coordinate systems, we can observe different distributions for this error estimation (*Figure 1D*). We propose that based on these distributions, specifically the variances of each distribution, the motor system can set the relative contribution of each coordinate system. One way of doing so is by assigning different weights to each coordinate system, for example, using an inverse variance estimation, which was shown to be an optimal way of combining different estimations to minimize the variance of the combined estimation (*Shadmehr and Mussa-Ivaldi, 2012*). The overall generalized forces are formed by multiplying the generalized forces of each representation with the respective weight.

## Results

We conducted four dynamic generalization experiments and showed that the generalized force patterns can be predicted by combining force patterns generated using the Cartesian, joint, and object coordinate systems using weights calculated from the level of noise in each coordinate system. In each experiment, participants learned to perform reaching movements while experiencing dynamic perturbations generated by a haptic robot (*Figure 2A*). While performing reaching movements between two points (*Figure 2B*), participants experienced velocity-dependent forces that were perpendicular to the movement direction and scaled according to it. That is, the forces could have different amplitudes and could have clockwise or counter-clockwise directions as a function of the motion direction (*Figure 2C*). A convenient way to representing this force field is to assign positive and negative signs to the clockwise and counter-clockwise directions, respectively. By doing so, the force pattern as a function of movement direction can be illustrated as a cosine wave (*Figure 2D*). After participants adapted to this force field at a training workspace, we tested the generalized forces pattern at two other spatial locations in the reachable space using a force channel technique (*Scheidt et al., 2001*). At these test workspaces, as well as in some trials at the training workspace, participants moved inside a channel, generated by virtual walls, which constrained their movement to a straight line (*Figure 2E*). While moving inside the channel, participants generated forces on the virtual walls that expressed their predictions of the force pattern that they could encounter while moving in this new workspace. Using these forces, we calculated the level of force compensation participants exhibited as a ratio between the produced and unscaled ideal forces (see Methods for more details).

In experiment 1, we used three arm configurations that set the training workspace, in which the participants experienced the scaled force field, and two test workspaces, in which we tested the force generalization pattern (*Figure 3A*). Participants adapted to the force field as indicated by both the reduction in maximum perpendicular error, measuring how much the force field perturbed the participants from moving in a straight line towards the target (*Figure 3B*), and the increase in predictive force compensation profile in the training workspace (*Figure 3C*, gray curve) that resembles the cosine tuning of the scaled force field (*Figure 2D*). The generalized force compensation in test workspaces 1 and 2 also exhibited a cosine pattern; however, it was shifted with respect to the curve of the training workspace or the curve representing the force field. For the curve calculated based on movements in the training workspace (*Figure 3C*, gray curve) we did not find a shift with respect to the curve of the force field (–0.01±0.03°, mean±STD). For the predictive force compensation curve in test workspace 1 (*Figure 3C*, blue curve), we found a leftward shift of –23.0±4.8° with respect to the force field curve. For the predictive force compensation curve in test workspace 2 (*Figure 3C*, red curve) we again found a leftward shift of the curve, however, this shift (–6.3±4.8°) was smaller compared with the shift we found for test workspace 1. In addition to the phase shift of the generalized force compensation pattern, we observed a reduction in the amplitude of the generalized forces. Based on previous studies such a decrease in generalized force magnitude might be due to the distance between the training and test workspaces (*Gandolfo et al., 1996*; *Berniker et al., 2014*).

These force compensation profiles can be predicted by a mixture model of the three coordinate systems while using the Re-Dyn-based weights. For each coordinate system, i.e., Cartesian, joint, and object, we calculated the predicted force compensation profile based on the arm configuration in the training and test workspaces. For test workspace 1, both the object-based and joint-based

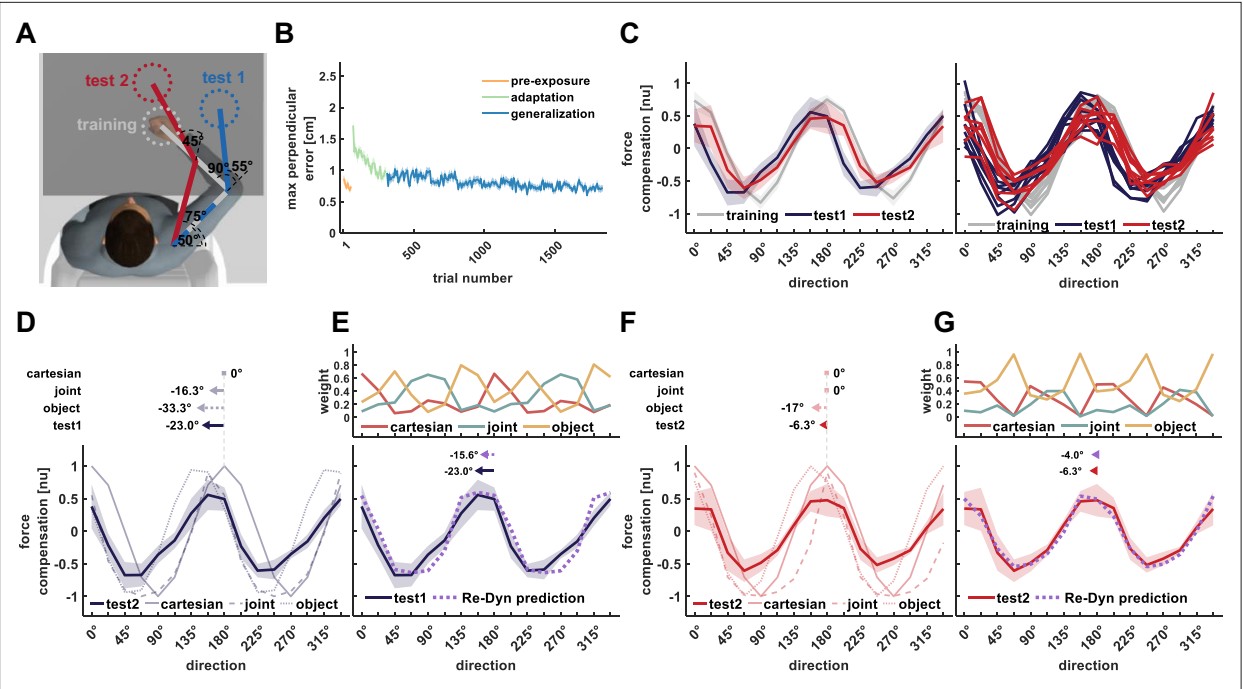

**Figure 3.** Results of experiment 1. (**A**) Workspace locations for experiment 1. Participants adapted to the scaled curl force field in the training workspace (gray) while force generalization was tested in test workspace 1 (blue) and test workspace 2 (red). Each workspace location was set based on predefined values of shoulder and elbow joint angles and the measured length of the upper and forearm. Only the start and target locations for the current movement were displayed on each trial. (**B**) Mean maximum perpendicular error and standard error (shaded area) across participants during the pre-exposure phase (orange line), adaptation phase (green line), and generalization phase (blue line) of movements performed in the training workspace. (**C**) Left panel, mean force compensation profiles. Participants were able to adapt to the scaled force field as evidenced by the force compensation profile in the training workspace (gray curve) and showed a leftward shift of the curve for test workspace 2 (red curve) and more so for test workspace 1 (blue curve). Right panel, individual force compensation profiles for each participant in the different workspaces. (**D**) Mean force compensation profile for workspace 1 (dark blue solid line) and Cartesian (light blue solid line), joint- (light blue dashed line), and object- (light blue dotted line) based model predictions. Arrows above the panel indicate the shift of the mean force compensation profile and the models' predictions with respect to the original curve describing the force field. (**E**) Upper panel, predicted weights for the Cartesian (red), joint- (turquoise), and object- (yellow) based representations for each movement direction according to the Re-Dyn model. Bottom panel, mean force compensation profile for workspace 1 (dark blue) and the predicted force compensation profile is generated by a mixture of reference frames according to the Re-Dyn predicted weights (purple). Arrows at the top indicate the shift of the mean force compensation profile and the model prediction. (**F**) Mean force compensation profile for workspace 2 (dark red solid line) and Cartesian (light red solid line), joint- (light red dashed line), and object- (light red dotted line) based model predictions. Arrows above the panel indicate the shift of the mean force compensation profile, similar to panel (**D**). (**G**) Upper panel, predicted weights for the mixture model, similar to panel (**E**). Bottom panel, mean force compensation profile for workspace 1 (dark red) and the Re-Dyn predicted force compensation profile (purple) similar to panel (**E**).

representations predict a leftward shift of the force compensation curve while the Cartesian-based representation predicts the same curve as the original curve describing the force field (*Figure 3D*). Importantly, none of the individual coordinate systems can accurately capture the experimental force compensation trend of the participants. However, when combining the three coordinate systems according to Re-Dyn-based weights (*Figure 3E*, upper panel), we could accurately predict this force compensation trend (*Figure 3E*, bottom panel). To scale the magnitude of the predicted force compensation pattern, we multiplied the weights by the decay factor that was equal to the peak-to-peak amplitude of the experimental force compensation pattern. By doing so, we kept the relative contribution of each coordinate system as predicted by the simulations. Despite the fact that the Re-Dyn prediction uses no information from the experimental results with the exception of the amplitude scaling, the prediction almost perfectly fits the complex pattern of forces produced by the experimental participants. Similar to test workspace 1, the Re-Dyn model could predict the combination between force profiles of the three coordinate systems that will produce the force compensation trend for test workspace 2. The individual predictions of each coordinate system for generalization in this workspace are depicted in *Figure 3F*. Similar to workspace 1, these individual predictions could not

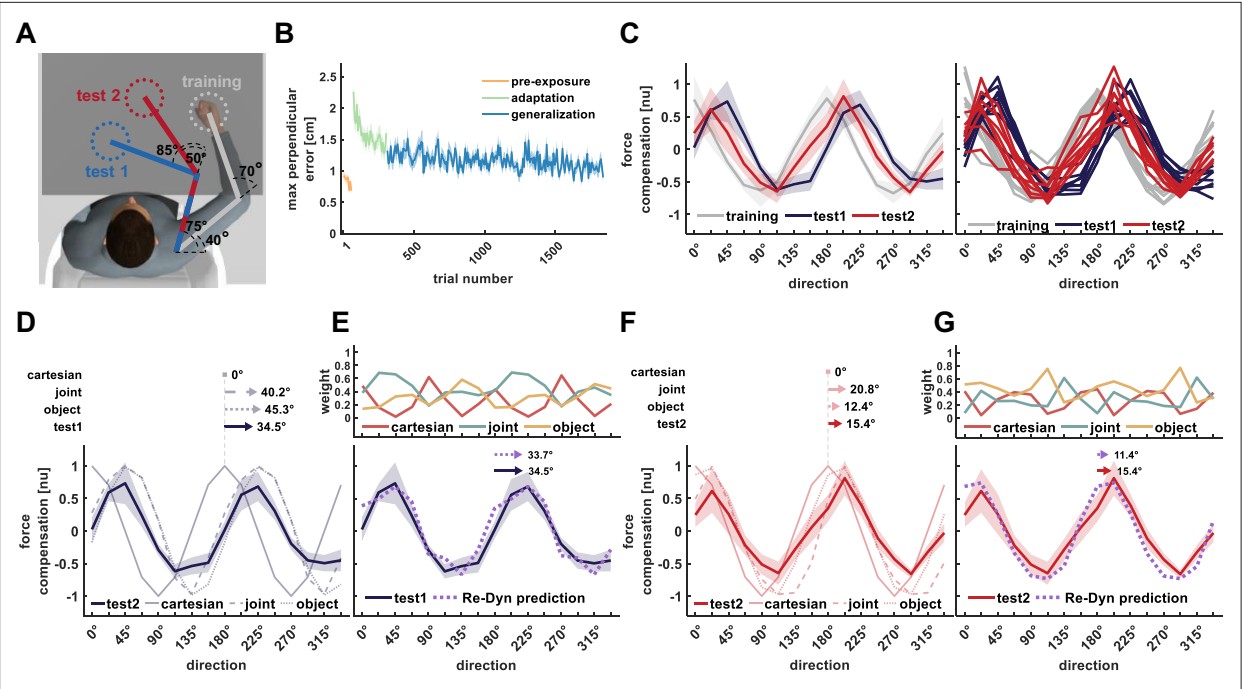

**Figure 4.** Results of experiment 2. All notations are similar to *Figure 3*. (**A**) Workspace locations for experiment 2. (**B**) Mean maximum perpendicular error (MPE). (**C**) Left panel, mean force compensation profiles in the training and test workspaces. Right panel, individual force compensation profiles in the training and test workspaces. (**D**) Mean force compensation profile for workspace 1 and model predictions (same as *Figure 3D*). Arrows above the panel indicate the shift of the mean force compensation profile and the models' predictions with respect to the original curve describing the force field. (**E**) Upper panel, the predicted weights for the mixture model for each movement direction according to the Re-Dyn model. Bottom panel, the resulted predicted force compensation profile generated by a mixture of reference frames (same as *Figure 3E*). Arrows at the top indicate the shift of the mean force compensation profile and the model prediction. (**F**) Mean force compensation profile for workspace 2 and model predictions (same as *Figure 3F*). Arrows above the panel indicate the shift of the mean force compensation profile and the models' predictions with respect to the original curve describing the force field. (**G**) Upper panel, the predicted weights according to the Re-Dyn model. Bottom panel, the resulted predicted force compensation profile is generated by a mixture of reference frames, the same as panel (**E**).

capture the experimental force compensation trend in workspace 2, while the combination of these reference frames according to the Re-Dyn weights predicts this trend almost perfectly (*Figure 3G*).

In experiment 2, we tested if the Re-Dyn model can predict generalized dynamics for a different set of arm configurations (*Figure 4A*). We found a similar adaption trend to the scaled curl force field as seen in the reduction in maximum perpendicular error (*Figure 4B*). The force compensation curve in the training workspace resembles the cosine tuning of the scaled force field (*Figure 4C*, gray curve), and we found a similar pattern for the force compensation in test workspaces 1 and 2 but for this experiment, they were shifted to the right with respect to the curve of the training workspace or the curve representing the force field. Similar to experiment 1, the predictive force compensation curve in the training workspace was not shifted with respect to the curve of the force field (–1±2.45°, mean ±STD). For the predictive force compensation curve in test workspace 1 (*Figure 4C*, blue curve), we found a rightward shift of 34.5±1.2° with respect to the actual force field. For the predictive force compensation curve in test workspace 2 (*Figure 4C*, red curve) we again found a significant rightward shift, however, this shift of 15.4±4.4° was smaller than that of test workspace 1.

Similar to experiment 1, we found that none of the individual coordinate reference frames were able to predict the shifts in the force compensation in the test workspaces (*Figure 4D and F*). However, again we found that the experimental force compensation on both test workspaces matched the Re-Dyn prediction. That is, these trends match the mixed model prediction based on weights calculated using the Re-Dyn model. Although the joint- and object-based representations predicted a rightward shift of the curve for test workspace 1 (*Figure 4D*) and a smaller rightward shift of the curve for test workspace 2 (*Figure 4F*), they did not fully account for the experimental force compensation curves. When combining these individual predictions according to the Re-Dyn weights, we found

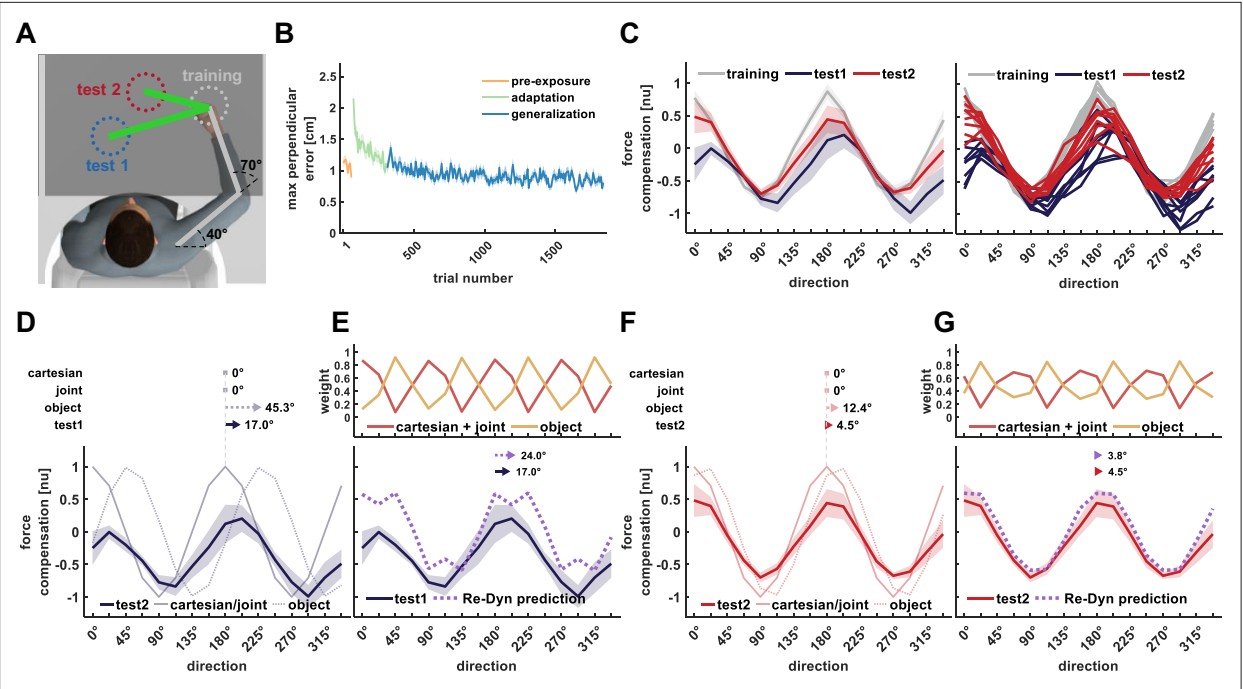

**Figure 5.** Results of experiment 3. All notations are similar to *Figure 3*. (**A**) Workspace locations. The participants always moved in the training workspace and the generalized forces were measured during movement of the endpoint of a pole (visually attached to the hand) in the same test workspaces as in experiment 2. (**B**) Mean maximum perpendicular error (MPE). (**C**) Left panel, mean force compensation profiles in the training and test workspaces. Right panel, individual force compensation profiles in the training and test workspaces (same as *Figure 3C*). (**D**) Mean force compensation profile for workspace 1 and model predictions (same as *Figure 3D*). Arrows above the panel indicate the shift of the mean force compensation profile and the models' predictions with respect to the original curve describing the force field. (**E**) Upper panel, the predicted weights for the mixture model for each movement direction according to the Re-Dyn model. In this case, since the Cartesian and joint-based generalized forces are identical, their weight is equal and thus we used a sum of the two weights (Cartesian +joint). Bottom panel, the resulted predicted force compensation profile is generated by a mixture of reference frames (same as *Figure 3E*). Arrows at the top indicate the shift of the mean force compensation profile and the model prediction. (**F**) Mean force compensation profile for workspace 2 and model predictions (same as *Figure 3F*). Arrows above the panel indicate the shift of the mean force compensation profile and the models' predictions with respect to the original curve describing the force field. (**G**) Upper panel, the predicted weights according to the Re-Dyn model. Bottom panel, the resulted predicted force compensation profile is generated by a mixture of reference frames, the same as panel (**E**).

an agreement between the predicted curve and the experimental curve for both test workspace 1 (*Figure 4E*) and test workspace 2 (*Figure 4G*). Importantly, we found a different pattern of weights in experiment 2 compared with experiment 1 which represents the changes in noise sensitivity due to the different arm configurations.

In experiment 3, we aimed to test whether the weighted combination of reference frames can predict generalized dynamics when arm configuration remains the same (*Figure 5A*). After adaptation to the scaled force field in the training workspace (*Figure 5B*), participants had to move the tip of a pole between the start and target positions in the test workspaces (*Figure 5A*). That is, the arm posture did not change between the adaptation and generalization parts of the experiment. We found a rightward shift of the curve for both test workspaces with respect to the training workspace. For the predictive force compensation curve in the training workspace (*Figure 5C*, gray curve) we did not find a shift with respect to the force field (–0.05±0.14°, mean±STD). However, for the force compensation curve in test workspace 1, we found a rightward shift of 17±5.7° (*Figure 5C*, blue curve) despite no changes in the arm posture. Aside from the phase shift, we also observed that the force compensation profile was not centered around the zero value as in the other experiments. In test workspace 2 we again found a rightward shift of the predictive force compensation curve, however, this shift (4.5±3.6°) was smaller compared with the shift we found for test workspace 1 (*Figure 5C*, red curve).

In this experiment, since the arm configuration did not change between the training and test workspaces, both the Cartesian and joint-based representations predict that the force compensation curve for the test workspaces will not shift compared with the force compensation curve for the training

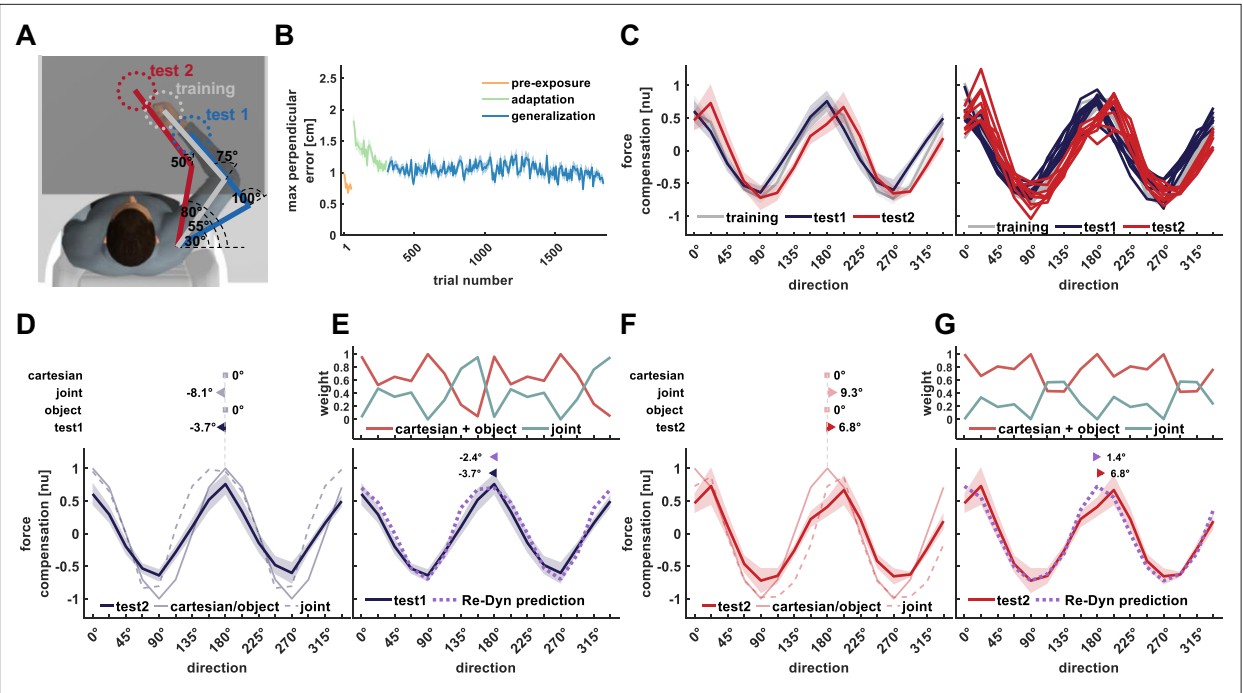

**Figure 6.** Results of experiment 4. All notations are similar to *Figure 3*. (**A**) Workspace locations for experiment 4. (**B**) Mean maximum perpendicular error (MPE). (**C**) Left panel, mean force compensation profiles in the training and test workspaces. Right panel, individual force compensation profiles in the training and test workspaces. (**D**) Mean force compensation profile for workspace 1 and model predictions (same as *Figure 3D*). Arrows above the panel indicate the shift of the mean force compensation profile and the models' predictions with respect to the original curve describing the force field. (**E**) Upper panel, the predicted weights for the mixture model for each movement direction according to the Re-Dyn model. Bottom panel, the resulted predicted force compensation profile generated by a mixture of reference frames (same as *Figure 3E*). Arrows at the top indicate the shift of the mean force compensation profile and the model prediction. (**F**) Mean force compensation profile for workspace 2 and model predictions (same as *Figure 3F*). Arrows above the panel indicate the shift of the mean force compensation profile and the models' predictions with respect to the original curve describing the force field. (**G**) Predicted weights the resulted predicted force compensation profile generated by a mixture of reference frames, same as panel (**E**).

workspace. Contrary to these predictions, due to the changed orientation of the tool, the object-based representation predicts a rightward shift for both workspace 1 and 2, with a stronger shift for workspace 1 (*Figure 5D and F*, dotted curves). For both test workspaces, we found that the predicted weights alternate between the strong contribution of the object-based coordinate system, the strong contribution of the Cartesian and joint-based coordinates systems, and a roughly similar contribution for all coordinate systems which depended on movement direction (*Figure 5E and G* upper panels). For test workspace 2, we observed a general agreement between the predicted and experimental force compensation curves (*Figure 5G*). However, for test workspace 1, since all individual models are centered around the zero compensation value, we could not capture the general amplitude shift of the force compensation curve (*Figure 5E*).

In experiment 4, we aimed to test the predicted combination between reference frames when the hand orientation remains the same between training and test workspaces while the arm configuration changes (*Figure 6A*). Similar to experiments 1 and 2, participants had to move at the test workspaces after adapting to the scaled force field in the training workspace. We found a similar adaptation to the scaled force field as evidenced by the reduction in MPE (*Figure 6B*). Again, the training workspace predictive force compensation curve (*Figure 6C*, gray curve) did not significantly shift with respect to the original force field curve (–0.03±0.07°, mean±STD). For test workspace 1, we found a small leftward shift of –3.7±3.1° (*Figure 6C*, blue curve) and for test workspace 2 we found a rightward shift of 6.8±1.8° (*Figure 6C*, red curve).

In this experiment, we set the location of the training and test workspaces in such a way that the hand orientation did not change across workspaces. In this case, both the Cartesian and object-based representations predict that the curves extracted from movements in the test workspaces will

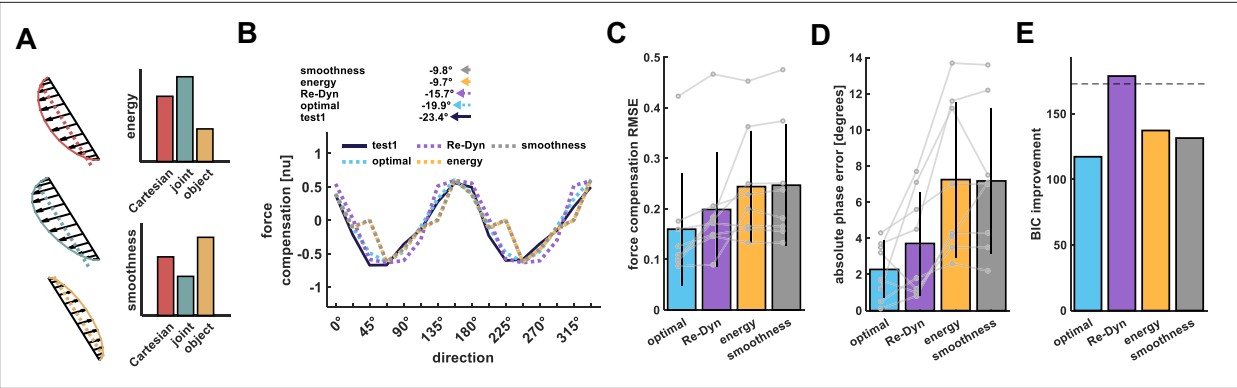

**Figure 7.** Comparison of prediction performance between Re-Dyn, energy, and smoothness-based models. (**A**) An example for combining the Cartesian, joint, and object reference frames force profiles using minimum energy or maximum smoothness criteria. Using the predicted force profiles of each reference frame we calculated the energy (dashed line) and smoothness of the signal envelope (solid line). In this example, the energy required to produce the generalized forces is the smallest for the object-based coordinate system (yellow) followed by the Cartesian (red) and joint (turquoise) coordinate systems. Similarly, the generalized force profile according to the object coordinate system is smoother compared with the other coordinate systems. Thus, in this example, the object-based coordinate system will have an elevated weight compared with the other coordinate systems according to both the minimum energy model and maximum smoothness model. (**B**) Example for the Re-Dyn (purple dashed line), energy (yellow dashed line), and smoothness- (gray dashed line) based models force compensation predictions for the experimental force compensation curve exhibited by participants in experiment 1 while moving in test workspace 1 (blue solid line). The weights between coordinate systems for the optimal fit (light blue dashed line) were calculated based on minimizing the error between the experimental curve and the fitted curve. Arrows at the top of the panel indicate the shift of the mean force compensation profile and the models' predictions with respect to the original curve describing the force field. (**C**) Bars represent the mean root mean square error (RMSE) between the predicted/fitted and experimental force compensation curves across all experiments. Error bars represent the standard deviation (n=8). Gray points represent individual RMSE values for each of the eight test workspaces we had in the four experiments. Lines connect the predictions/fitting for each test workspace. (**D**) Same notation as in C but for the mean phase shift error between the experimental force compensation profiles and the phase shift of the models' predictions. (**E**) BIC Improvement for each of the models relative to no generalization (that is, a model in which the force is zero at all movement directions). Dashed line shows the cutoff for models that are not considered distinguishable in terms of their performance from the best model (Re-Dyn model).

resemble the ones extracted from the training workspace. Contrary to these predictions, the joint-based representation predicts a leftward shift for the force compensation curve extracted from test workspaces 1 and a rightward shift for the curve extracted from test workspaces 2 (*Figure 6D and F*). We found that in general, the Cartesian and object-based representations are the more dominant reference frames as evidenced by the predicted weights (*Figure 6E and F*). For most directions, the weight for these reference frames is higher than the weight for the joint reference frame, although for some directions this gap in weight value is small and could even reverse. This contribution of the joint-base reference frame explains the small phase shift of the mixture model that matches the experimental results.

In addition to the noise sensitivity model we considered two alternative mechanisms to explain the weighting between reference frames. The first suggests that the weights are set by minimizing the energy needed to generate the generalized forces. For every movement direction, the motor system will give more weight to the reference frame in which the applied forces, which are proportional to the energy, are the lowest. Similar to the Re-Dyn model, we used normalized inverse energy measurement to calculate the weight of each reference frame for the four experiments (*Figure 7A*). The second alternative mechanism suggests that the motor system is trying to maximize the smoothness of the force profile. Similar to other studies that suggested smoothness as a fundamental, target aim for the motor system, we tested whether maximizing the smoothness of the generalized force profile may set the weighting between reference frames. For this purpose, we calculated the smoothness, measured by the inverse sum of the magnitude of the force signal third derivative (see Methods), and set a higher weight for more smooth force profiles (*Figure 7A*).

An example for the predicted force compensation profile of these two alternatives for the mean force compensation of participants in experiment 1 is depicted in *Figure 7B*. Importantly, none of these three models were directly fitted to the experimental data. To test the prediction quality of the three models, that is the Re-Dyn, energy, and smoothness-based models, we also found an optimal force compensation profile by fitting the weights of the three reference frames directly to the data.

The weights were independent of the movement direction and fitted to all three reference frames since this combination was superior to all other combinations (see Appendix 1 for comparison with other combinations). That is, we found a set of three weights that combine the reference frames so it minimizes the error between the fitted and experimental force combination profile. This fitted curve quantifies how well each of these models can predict the results in comparison with a fitted optimal constant mixture of the Cartesian, joint, and object reference frames. Note that the optimal model is directly fitted to the data (unlike the other model predictions), and provides an upper bound, demonstrating how close any of the predictions are to a directly optimized model (for comparison). In this example, we observed that both the force compensation magnitude and the phase shift are better predicted by the Re-Dyn model compared with the energy or smoothness-based models. Across the eight generalization test workspaces we had in the four experiments, we found that the Re-Dyn model has less prediction error, as measured by the error between the prediction and experimental force compensation curves (*Figure 7C*), and better predicts the phase shift of the force compensation curve (*Figure 7D*). As expected, the predictions of the Re-Dyn model were not as good as the optimal weights model that was fitted to the data. This is due to the inherent additional free parameters in the optimal model. To account for the additional degrees of freedom, we calculated the Bayesian Information Criterion (BIC) for each model. We observed that the Re-Dyn was superior to the other models, including the fitted optimal model, when accounting for the additional degrees of freedom (*Figure 7E*). In addition, we found an additional condition in which the direction-independent optimal weights model cannot predict the experimental results despite the additional parameters (see Appendix 2). In general, these results suggest that the Re-Dyn model better captures the weighting between coordinate frames compared to all alternative models.

Our results strongly suggest that the motor system alternates between different reference frames according to their reliability. To further test the ability of the Re-Dyn combination of reference frames to predict dynamic generalization, we examined whether this model can also explain past experimental results. Previous studies did not provide a consistent view regarding the way the motor system represents dynamics. That is, each of these studies provided evidence for representation using the Cartesian, joint, and object-based systems or some mixture of them. Our simulations examined whether this inconsistency is a result of the experimental design that was used, which elevated the noise level in one coordinate system resulting in the motor system using other coordinate systems.

First, we examined the study by *Shadmehr and Mussa-Ivaldi, 1994*. Similar to the experiments reported here, the authors examined the generalized dynamics of participants in a test workspace after adaptation to a skew viscous force field in the training workspace (*Figure 8A*). To understand how the brain represents the learned forces, the authors examined the movement profiles in the test workspace in which participants did not experience any forces, i.e., moving in free space. In this case, the movement path will divert from the straight line due to the additional counteractive forces that participants generate to overcome anticipated external forces. Therefore, using different reference frames to represent the external dynamics will result in different movement paths during these free-space movements and the trajectories can reveal the reference frame the motor system used during movements. Based on these paths, the authors suggested that the motor system used the joint coordinate system to represent novel dynamics. Simulating this experimental protocol, we found that the force pattern of the joint and object-based coordinate system is similar (*Figure 8B*) and that the Re-Dyn predicted weights of these coordinate systems have a higher value compared with the weight of the Cartesian coordinate system (*Figure 8C*). Since the authors did not consider the object-based coordinate system, they concluded that the observed generalization is a result of the motor system using the joint-based coordinate system, although it is impossible to differentiate between the joint or object coordinate systems based on this design. We used the predicted weights, including an object-based representation, and simulated the free-space movement at the test workspace after adaptation to the force field in the training workspace (*Figure 8D*). We found that the simulation results, specifically the direction of deviation from the straight-line path, can capture the experimental results (*Figure 8E*).

Other influential studies also supported the conclusion of *Shadmehr and Mussa-Ivaldi, 1994*. For example, *Shadmehr and Moussavi, 2000* and *Malfait et al., 2005* examined dynamic generalization using a similar protocol in which participants adapted to a force field in the training workspace and generalization was tested in a different workspace location. For both studies, our simulations of the

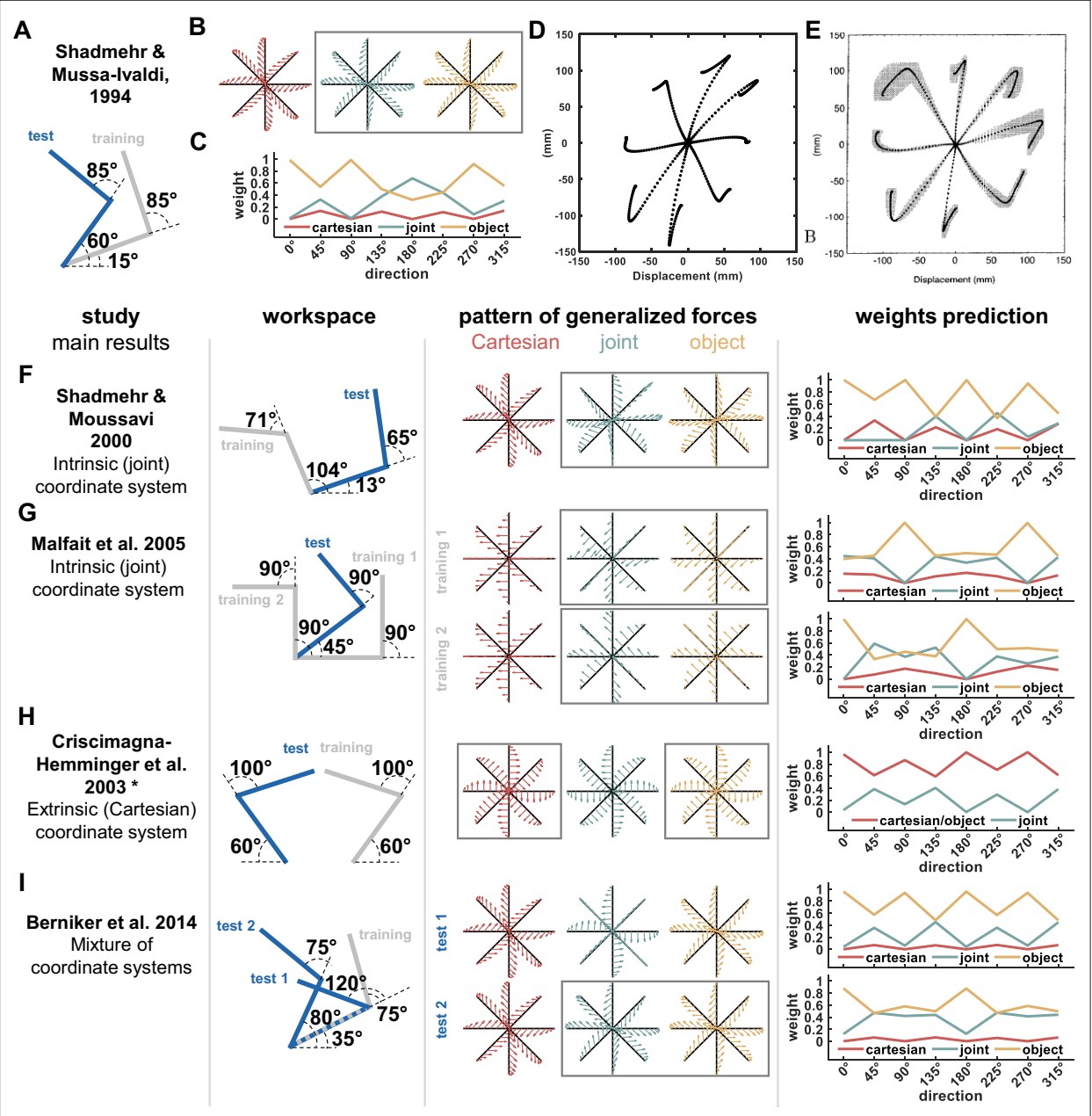

**Figure 8.** Predictions of the Re-Dyn model for weighting between coordinate systems for previous studies. (**A**) Workspaces locations for the study by *Shadmehr and Mussa-Ivaldi, 1994*. (**B**) Pattern of generalized forces for the Cartesian (red), joint (turquoise), and object (yellow) coordinate systems. Arrows mark the direction and magnitude of each coordinate system predicted forces after adapting to a skew force field that was used in the original study. In this case, the joint and object-based coordinate system predict similar pattern of generalized forces (marked using a gray rectangle). (**C**) For the workspaces in (**A**), we calculated the Re-Dyn predicted weights between coordinate systems used for dynamic generalization. (**D**) We used an internal model which generates movements in free space after adaptation to the skew force field similar to the internal model used in the original study. In this case, since there are no external forces, we observed the after-effect movement pattern, i.e., movements that deviate from the straight line path due to counteractive forces produced by the internal model. The internal model uses a weighted sum of the three coordinate systems, Cartesian, joint, and object to generate these forces according to the weights in (**C**). Dashed lines represent paths to eight equally spaced targets from a center initial point. (**E**) Experimental results of the after-effect movements in the test workspace. The simulation results in (**D**) capture the deviation direction from the straight-line path, explaining how different coordinate systems were combined in this experiment. Moreover, since the joint and object coordinate systems predict the same pattern of generalized movement as well as they are the dominant systems in this representation, it can explain why the authors of the original study concluded that dynamic generalization is achieved using the joint coordinate system. Figure 8E is reproduced from Figure 14B from *Shadmehr and Mussa-Ivaldi, 1994*. (**F–I**) Predictions of the dominant reference frame according to the Re-Dyn model for previous experimental designs and comparison of these predictions with the conclusions of these past studies. Each row of panels includes the information

*Figure 8 continued on next page*

*Figure 8 continued*

presented in panels A-C for different studies. From left to right, panels present the main result of the study regarding the dominant coordinate system for dynamic representation, hand configuration used in the study setting the training and test workspaces locations, predicted generalized force patterns, and the simulated weights according to our Re-Dyn model. For the last panel, the simulated weights for Cartesian, joint, and object-based representations are marked using red, turquoise, and yellow lines, respectively. * In the original paper the authors did not mention the exact shoulder and elbow joints values, but instead mentioned that the arms were symmetric about the midline and workspaces were close to the midline. We estimated the configuration based on the sketch in Figure 1 in the original paper.

generalized force pattern showed great similarity between the object and joint coordinate systems and that Re-Dyn predicted weights support that these coordinate systems have higher values compared with the Cartesian reference frame suggesting that participants will rely more on these force representations (*Figure 8F and G*). Similar to *Shadmehr and Mussa-Ivaldi, 1994*, the object reference frame was not considered by the authors of the two studies which explains why they concluded that representation is based on the joint-based coordinate system.

Next, we examined predictions of the Re-Dyn model for studies that had different conclusions to *Shadmehr and Mussa-Ivaldi, 1994*. In the study of *Criscimagna-Hemminger et al., 2003* the authors tested generalization between hands. Participants who adapted to a force field while using their dominant hand generalized these forces to the non-dominant hand and the force profile supported representation using the Cartesian reference frame. The authors used a mirror configuration for the hands (*Figure 8H*), that is, the training and test workspaces were at the same spatial location. In this case, the generalized force profiles predictions for both Cartesian and object-based representations are the same. According to our simulations for the Re-Dyn predicted weights, we found that the weights for the Cartesian and object-based representations are indeed higher than the weights for the joint-based representation which explain the reason the authors concluded that participants used the Cartesian coordinate system for inter-limb dynamic generalization.

Another experimental result that can be explained using the Re-Dyn model suggested a combination of reference frames. In the study by *Berniker et al., 2014*, the authors suggested that generalized forces after adaptation to skew force field are a result of combining the Cartesian, joint, and object-based coordinates systems. Simulations of the experimental setup (*Figure 8I*) explain this result as we found that for one of the tested configurations, the generalized force patterns of the joint and object coordinate system are similar while for another tested workspace, the generalized force pattern of the joint coordinate system is the mean of the Cartesian and object force patterns. These links between the force patterns did not allow the authors to clearly indicate which reference frame is used by the motor system. This is supported by the Re-Dyn model since the combination between reference frames according to the predicted weights generates a force pattern that can be interpreted using multiple and different combinations of reference frames.

## Discussion

In this study, we showed that during dynamic generalization the motor system alternates between different reference frames according to their reliability. Since motor commands sent to the muscles are corrupted by neural noise, there is an increased risk of missing the movement goal. For example, during reaching movement in the presence of external force perturbations, producing inaccurate counteractive forces can result in missing the movement target. To overcome this error, we suggest that the motor system chooses to control these forces by representing and producing forces according to a coordinate system in which it can lower the effect of neural noise. To support this theory, we compared between the Re-Dyn model predicted generalized force patterns and experimental patterns outside the training workspace where participants adapted to a scaled curl force field. The scaling of the force field allowed to compare both the error in force magnitude and the general phase shift of force pattern. We found that the generalized forces resembled the prediction of a mixture model combining Cartesian, joint- and object-based representations. Importantly, we show that the weight for each coordinate system was best predicted by a model that asserts that the weights are inversely proportional to the force error produced when using a specific coordinate system. Importantly, these

Re-Dyn model weights were directly determined from the experimental conditions and were not fit to the participant data. Thus, each experiment provides a direct test of the model predictions. Moreover, we further show that this way of calculating weights between coordinate systems can solve the inconsistent results of previous studies examining dynamic representation in the motor system.

Our idea suggests that the motor system uses a mix of force representation with a greater weight to the representation that is less variable in extrapolated force production. While the idea of trajectory planning while minimizing the effect of noise was presented by *Harris and Wolpert, 1998*, we show that the motor system alternates between controlled variables, that is, end-point, joints, or object-based forces, in order to reduce the effect of noise aside from optimizing the trajectory. By doing so, the motor system can ensure that even in the case of elevated noise, the effect will be kept to a minimum.

We focused on additive noise while the real noise in the motor system is more likely to be signal-dependent noise (*Jones et al., 2002*; *Todorov and Jordan, 2002*). For signal-dependent noise, the noise characteristics change as a function of the forces the muscles need to produce. In this case, since different reference frames predict different levels of force for different movement directions, elevated noise can increase the planned forces corruption of one reference frame compared to others, changing the weighting between coordinate systems. While this can result in better resemblances of the model prediction to the experimental data as the number of degrees of freedom for the model increases, we found that even when keeping the same noise characteristics across reference frames and movement directions we could explain the generalization trends in multiple cases.

We used simulations of neural noise added to different state variables that define the controlled forces to estimate the error between desired and corrupted force output during generalization. Using this error distribution, we calculated the weight for each reference frame based on the inverse variance estimation. While we show that this weight estimation can predict the force compensation patterns under different conditions, our study does not aim to support the inverse variance framework. Indeed, there is an ongoing debate regarding the relevance of the inverse variance in perception and motor control (*Rahnev and Denison, 2018*). In contrast, we aimed to show that the motor system takes into consideration the possible errors in force representation using different coordinate systems.

We expect that these representations and related weights might be stored in areas related to forward and inverse models. During motor adaptation, the sensorimotor system might change both the controller (inverse model) and the predictor (forward model) to modify the control so as to overcome changes in the environment (*Flanagan et al., 2003*; *Franklin et al., 2008*). The cerebellum has been proposed to host both types of internal models (*Wolpert and Kawato, 1998*; *Smith and Shadmehr, 2005*), however, other areas might also be involved during adaption to altered environment dynamics. For example, changes in premotor cortex (PMd) activity were also evident during force field adaptation (*Perich et al., 2018*). Generalization of the learned dynamics to different locations in space might require both the inverse and forward models to use dynamics transformation such as the ones considered in this study. Thus, we speculate that such transformation might be evident in the neural activity of these areas during adaptation to novel dynamics.

The results we present here support the idea that the motor system can use multiple representations during adaptation to novel dynamics. Specifically, we suggested that we combine three types of coordinate systems, where each is independent of the other. Other combinations that include a single or two coordinate system can explain some of the results but not all of them, suggesting that force representation relies on all three with specific weights that change between generalization scenarios. However, we can also consider that force is represented in other coordinate systems. For example, when we reach to a target, we need to represent the location of the target in eyes/head coordinate system and transform this representation through body/world-based coordinate system to hand coordinate system (*Flanders et al., 1992*; *Andersen et al., 1993*; *Buneo et al., 2002*). While we considered body/world (Cartesian) and hand (joint) coordinate systems, the gaze-based coordinate system could also be a potential coordinate system in which we can represent forces. However, since in our experiments, participants were not required to gaze at a specific point in the environment and usually were gazing at the cursor, the sagittal axis orthogonal to the center of this coordinate system was always aimed at the workspace in which the participants were moving. This will result in no change in the generalized force profile, similar to the Cartesian-based coordinate system prediction. Alternatively, making participants gaze at a specific point will result in a generalized force profile that is similar

to the object-based coordinate system. That is, since the difference in location between workspaces on the retina can be described using a rotation operation, the predictions in such case will be similar to the object coordinate system which is based on the same operation. Thus, while in both cases of fixed or free gaze, we cannot distinguish between the gaze and other coordinate systems, the identical predictions of these coordinate systems were still taken into consideration in the Re-Dyn model.

A different approach to try and examine the possible representations that are present in the motor system can be to try and limit the use of some of them during adaptation to novel dynamics. Since the force field is ambiguous (i.e. the force field appeared in a small workspace with no constraints on which coordinate frame to use for representation of the dynamics) it can be represented in any of the suggested coordinate frames including a mixture of them. In this case, the motor system extrapolates the pattern of forces to other locations in space and based on our suggestion it does that using the most reliable reference frame. However, this can change in the case of unambiguous force fields that extend over a larger area of the workspace according to one coordinate frame (*Franklin et al., 2016*). Adaptation to unambiguous force fields might result in using a single coordinate frame for dynamics representation since the prediction of forces is based on interpolation rather than extrapolation.

Force generalization has been an important way to test the coordinate system the motor system uses to represent information (*Franklin and Wolpert, 2011*). The initial idea that the motor system uses intrinsic, joint-based representation (*Shadmehr and Mussa-Ivaldi, 1994*) is more questionable nowadays as more evidence accumulated showing that the motor system can use other coordinates systems (*Saha et al., 2015*) as well as using a mixture of models (*Brayanov et al., 2012*; *Berniker et al., 2014*). We propose that these discrepancies suggest different weighting between reference frames based on their reliability rather than a direction-independent reference frame that is used for all cases of dynamics representation. As we show, the Re-Dyn framework can explain generalization between different locations and types of force fields as well as other types of generalization tests, such as inter-limb transfer (*Criscimagna-Hemminger et al., 2003*), and assist in characterizing possible failure of information representation, such as brain-related disorders (*Haswell et al., 2009*).

## Methods

### Participants

Thirty-two right-handed participants (14 females; aged between 22–35) participated in one of four experiments (eight participants per experiment) after signing a consent form (that included consent for participating in the study and consent for data processing and publication) and completing a handedness test (*Oldfield, 1971*). The experimental protocol was approved by the Ethics Committee of the Medical Faculty of the Technical University of Munich (number 763/20 S-KH). Each participant only performed one of the four experiments.

### Experimental setup

We created a haptic augmented virtual reality environment using a vBOT robotic device (*Howard et al., 2009*). The robotic device was used to record hand position and velocity while generating force feedback at 1 kHz in real-time. A six-axis force transducer (ATI Nano 25; ATI Industrial Automation) measured the end-point forces applied by the participant on the handle and was also sampled at 1 kHz. Participants were seated in front of the experimental setup and looked at a semi-silvered mirror showing the projection of an LCD screen placed directly above it (*Figure 2A*) such that visual feedback was presented in the plane of arm movement. An opaque screen below the mirror prevented participants from viewing their hand. The participant grasped the robotic handle using his/her right hand with their arm supported on an air sled allowing frictionless movement. Once the participant was seated in front of the system, we measured the shoulder position in system coordinates. To do so, we initially measured the upper arm length and the length between the elbow joint to the handle center. Next, we placed the hand at the system center location by applying forces using the robot and measured the elbow and shoulder angles. Using these measurements and the inverse kinematics equations of a two-link arm model, we calculated the shoulder location. During the experiment, we assume participants maintain a fixed wrist angle and thus we did not fix their wrist joints similar to previous studies (*Shadmehr and Mussa-Ivaldi, 1994*; *Shadmehr and Moussavi, 2000*; *Malfait et al., 2002*; *Malfait et al., 2005*). The wrist deviation during force field and generalization experiments was

shown to be small and thus have a negligible effect (*Berniker et al., 2014*). Participants performed point-to-point reaching movements between a start position and a target position. The participants saw on the screen a cursor (red circle with a diameter of 0.7 cm) representing their hand. Each trial was started by the robotic system moving the participant's hand to the start position (1.5 cm diameter gray circle) in which the participants were instructed to wait for a go cue. After a random wait time varying between 0.75 and 1.5 s, a short auditory beep signaled participants to start their movement towards the target (yellow circle with 1.5 cm diameter). The start and target points were chosen from 16 equally spaced locations on a 10 cm diameter circle (*Figure 2B*). Participants were instructed to move with a consistent peak speed of 50 cm/s. To do so, at the end of each movement participants were presented with visual feedback about the peak speed of their movement. Peak speed that was within ±8 cm/s from the desired peak speed was considered 'good' while a peak speed below this value received a 'too slow' message and above it received a 'too fast' message. In addition, if the participants overshoot the target position by more than 2 cm, they received an 'overshoot' message. For each movement that was considered as a good movement, we increased a score counter that appeared on the top of the screen by one point.

On each trial, the robot was used to produce one of three virtual environments. The first was a null field (NF), in which the robot was completely passive. For the other two environments, the robot generated forces on the participants' hand during movement. The second virtual environment was a *force channel* (*Scheidt et al., 2001*; *Milner and Franklin, 2005*; *Smith et al., 2006*), on which we implemented two virtual walls that constrained the participants' hand to move in a straight-line path to the target. The virtual walls were created by generating forces that were perpendicular to the line connecting the start and target positions. The forces were calculated according to the deviation of the hand position from the straight-line path multiplied by a stiffness coefficient of 4000 N/m and the velocity in the movement perpendicular direction multiplied by a damping coefficient of 2 Ns/m. The third type of virtual environment was a *force field*. During these trials, participants experienced forces acting on their hand that were a function of the hand's instantaneous velocity. The forces were calculated using a scaled curl matrix.

$$\begin{bmatrix} F_x \\ F_y \end{bmatrix} = B\left(\dot{x}, \dot{y}\right) \cdot \begin{bmatrix} \dot{x} \\ \dot{y} \end{bmatrix} = \begin{bmatrix} 0 & 0.16 \\ -0.16 & 0 \end{bmatrix} \cdot cos\left(2 \cdot arctan\left(\frac{\dot{y}}{\dot{x}}\right)\right) \cdot \begin{bmatrix} \dot{x} \\ \dot{y} \end{bmatrix} \qquad (1)$$

where $F_x$ and $F_y$ are the forces generated in the x and y axis, respectively and $\dot{x}$ and $\dot{y}$ are the hand velocities in the x and y axis, respectively. The traditional curl force field was scaled using a cosine trigonometric function of movement direction. That is, the magnitude and sign of the scaling factor changed according to movement direction, calculated using the inverse tangent function (*Figure 2C*). For example, when the motion was made only in the x-axis, i.e., horizontal right to left or vice-a-versa, the scaling factor was equal to 1, or when the motion was made only in the y-axis, i.e., moving away or towards the body, the scaling factor was equal to (–1). We define clockwise forces as positive forces and counter-clockwise forces as negative forces. Simulated mean forces that participants experienced can be illustrated as a cosine curve when plotted as a function of movement direction (*Figure 2D*).

## Experimental protocol

We conducted four experiments using similar experimental protocols. For each experiment, we used three workspaces (differing in their spatial location) in which participants performed the reaching movements; one *training workspace* and two *test workspaces*. Each experiment included four experimental phases. The experiment started with a familiarization phase of 16 trials that was performed only in the training workspace. In this phase, participants performed reaching movements from each one of the sixteen starting points. The movements were unconstrained reaching movements, i.e., there were no forces applied to participants' hand (NF condition). The second, pre-exposure phase included recording baseline measurements of the movements in the force channel. For this purpose, participants performed 384 movements overall in the training and test workspaces. The movements included both unconstrained reaching movements (NF condition) and movements while the robot generated a virtual force channel between start and target locations (FC condition). Participants performed 4 NF and 4 FC trials for each of the sixteen starting points in each workspace, which means that for each workspace participants performed 128 movements (a total of 384 movements across

the three workspaces). For these trials, we randomized the start point and type of condition using a uniform distribution. We divided the 384 into four blocks of trials and randomized the appearance of the condition, that is NF or FC, and the sixteen starting points for each of the three workspaces using a uniform distribution within a block. During the third, exposure phase, participants experienced the forces generated by the trigonometric-scaled curl force field (FF condition). Participants performed 15 repetitions from each of the 16 starting points (total of 240 trials), all performed in the training workspace. We divided the 240 trials into fifteen blocks in which each of the starting points appeared once and the order of appearance within a block was drawn using a uniform distribution. Finally, after adaptation to the force field, the fourth phase tested generalization of the force field representation in the two test workspaces. During this phase, participants performed 1536 reaching movements in the training workspace on which the force field was applied and an additional 384 movements in the training and test workspaces in which their movements were constrained within the force channel. For each workspace, we had eight repetitions of force channel trials for each of the 16 starting points (a total of 384 trials). These trials were randomly introduced within the 1536 force field trials; for every series of four force field trials, we had one force channel trial. The position of the force channel trial within the series was random with a uniform distribution. Each participant participated in one of four experiments, which differed in the location of the training and test workspaces or had an additional task besides performing reaching movements.

### Experiment 1
The location of the center of the training workspace was set for each participant so that the participants' shoulder joint was at 50° and elbow joint was at 90°. The center of test workspace 1 was set so with participants' shoulder angle at 50° and elbow angle at 55°. The center of test workspace 2 was set with the shoulder joint at 75° and the elbow at 45° (*Figure 3A*).

### Experiment 2
The location of the center of the training workspace was set for each participant so that the participants' shoulder joint was at 40° and elbow joint was at 70°. The center of test workspace 1 was set so with participants' shoulder angle at 75° and elbow angle at 85°. The center of test workspace 2 was set with the shoulder joint at 75° and the elbow at 50° (*Figure 4A*).

### Experiment 3
The location of the training and test workspaces were set specifically for each participant and identical to those of experiment 2. However, in this experiment, the participant's hand always moved in the training workspace (both training and test workspaces). Movements in the test workspaces were performed using a virtual pole attached to the participants' hand. That is, a virtual green pole linked the red cursor representing the participants' hand to a green cursor located in the test workspace. This green cursor then moved with this fixed offset to the participant's hand, as the participant's hand moved. We instructed participants to move the green circle between the start and target instead of their hand. The length and orientation of the pole were set based on the vector connecting the center of the training and test workspaces (*Figure 5A*). To move between a particular set of start and target points in the test workspace, participants had to perform similar movement between the same start and target points in the training workspace with their hand. Thus, the joint trajectory was similar regardless of whether it was performed in the training or test workspaces.

### Experiment 4
This experiment was similar to experiments 1 and 2, but with different workspace locations (*Figure 6A*). The center of the training workspace was set for each participant so that their shoulder joint was at 55° and elbow joint was at 75°. The center of test workspaces 1 and 2 was set such that the shoulder and elbow joints were oriented at [30°, 100°] and [80°, 50°], respectively. These workspace locations kept the hand orientation constant while the arm configuration varied.

## Data analysis
Data analysis was performed offline using MathWorks MATLAB (RRID:SCR_001622, v.9.5).

## Maximum perpendicular error (MPE)

On each null field or force field trial, the maximum distance between the trajectory and the straight line joining the start and target points was calculated as a measure of kinematic error.

## Force compensation

The force compensation was calculated during the generalization phase for the training and test workspaces on the force channel trials. Force compensation in the training workspace indicates the degree of adaptation of individual participants to the force field whereas force compensation in the test workspaces demonstrates how the learned forces are generalized in space. Force compensation was calculated as the regression slope between the perfect force compensation signal, the independent variable, and the recorded force signal, the dependent variable (*Smith et al., 2006*). The recorded force profile for each channel trial was identified between the start and end of the movement (first and last point at which velocity exceeded 5% maximum velocity). The perfect compensation force profile was determined as the force required for perfect compensation of movement in an unscaled curl force field and was calculated by multiplying the unscaled curl force field matrix by the velocity vector, $\begin{bmatrix} 0 & 0.16 & ; & -0.16 & 0 \end{bmatrix} \cdot [\dot{x}; \dot{y}]$. Here, we compare the recorded force profile with forces computed using an unscaled version of the curl force field and not the scaled curl force field for three reasons. First, for some directions there were no force perturbations (e.g. moving at a direction of 45°), even the smallest force participants generate during force channel trials will result in an infinite slope value. The second reason is to capture the direction of the forces that the participant generated, i.e., clockwise or counter-clockwise. Regression against always positive values of the perfect force profile allows us to identify in which directions the participants generated positive or negative forces. This allows us to identify any shift in the force compensation profiles when plotted as a function of movement direction. Third, by calculating the regression against the same value across all directions allows us to identify the force magnitude participants adapted to as a function of movement direction. Once we calculated the force compensation measurement for each movement in the force channel, we averaged across the eight repetitions for each movement direction and repeated this for all 16 directions and each workspace. The resulting pattern of the force compensation measure across all movement directions resembles a cosine function. To quantify phase differences between the scaled force field and the force compensation curves extracted from movements in the training and test workspaces, we calculated the cross-correlation between these cosine signals. That is, the cross-correlation provided a measure for the shift of each force compensation curve from the designed scaled force field.

## Prediction of generalization models

We considered three different coordinate frames in which the motor system could represent the learned dynamics. The first coordinate system we considered is the Cartesian coordinate system. For this representation, the forces are invariant with respect to the arm orientation. In such case, participants will exhibit similar forces to the learned forces for each one of the test workspaces (*Equation 1*), resulting in a similar cosine signal of generalized forces as a function of movement direction.

The second coordinate system we considered is the joint-based coordinate system (*Shadmehr and Mussa-Ivaldi, 1994*; *Malfait et al., 2002*). For this representation, the generalized forces exhibited in the test workspaces should differ from the learned forces since the arm configuration has changed. That is, for experiments 1, 2, and 4 we expect a change in the cosine function describing the generalized forces. To calculate the expected generalized forces profile, we used a two-link arm model in which $\theta = [\theta_s, \theta_e]$ represents a vector of the shoulder and elbow joints values. Initially, we need to derive the learned dynamics in joint coordinate system and then derive the expected generalized forces. In joint representation, the learned forces are represented as joints torques using the Jacobian which forms the relation between the two variables, $\tau = J_{train}^T \cdot F$, where $F = [F_x, F_y]$ is a vector of the end-point forces, $\tau = [\tau_s, \tau_e]$ is a vector of joints torques, and $J_{train}$ is the arm's Jacobian matrix for the training configuration. Similarly, the hand velocities are represented using joint angular velocity and the Jacobian matrix, $v = J_{train} \cdot \dot{\theta}$, where $v = [\dot{x}, \dot{y}]$ is the hand velocity vector. Thus, using the end-point forces generated by the force field (*Equation 1*) we derived the learned joint torques.

$$\tau = J_{train}^T \cdot F = J_{train}^T \cdot B(v) \cdot v = J_{train}^T \cdot B\left(J_{train} \cdot \dot{\theta}\right) \cdot J_{train} \cdot \dot{\theta} \tag{2}$$

The expected forces during movement in the test workspaces can be computed using the learned torques and the arm's Jacobian for the test configuration.

$$
\begin{aligned}
F &= J_{test}^{-T} \cdot \tau \\
&= J_{test}^{-T} \cdot J_{train}^{T} \cdot B\left(J_{train} \cdot \dot{\theta}\right) \cdot J_{train} \cdot \dot{\theta} \\
&= J_{test}^{-T} \cdot J_{train}^{T} \cdot B\left(J_{train} \cdot J_{test}^{-1} \cdot v\right) \cdot J_{train} \cdot J_{test}^{-1} \cdot v
\end{aligned}
\tag{3}
$$

The third coordinate system we considered is the object-based coordinate system (*Berniker et al., 2014*). For this representation, the motor system relates the dynamics to a grasped object. When moving outside of the training workspace, the expected forces are calculated according to the orientation of the held object, which is rotated with the hand orientation. For example, if we learned a force profile while moving straight away from the body and we now moved to another location in which the hand was rotated by an angle $\theta$ in external space, the learned profile is expected to appear in a direction that is rotated by $\theta$. That is, initially, based on the velocity vector, we can find the learned force profile by rotating back to the original training workspace $F_{train} = B\left(R^{-1}\left(\theta\right) \cdot v\right) \cdot R^{-1}\left(\theta\right) \cdot v$, where R is a rotation matrix. Then, we rotate these forces to the test workspace:

$$
F_{test} = R\left(\theta\right) \cdot F_{train}
\tag{4}
$$

We simulated the expected forces according to the models for the two test workspaces using the different arm postures. For experiment 1 (*Figure 3D and F*), using a Cartesian-based representation, the expected forces do not change when switching between workspaces. Thus, the force compensation profile is similar to the profile of the force field for both test workspaces 1 and 2. In the case of a joint-based representation, since the arm posture changes between workspaces, the force compensation profile for the test workspace changes compared with the learned force field. For test workspace 1, there is a leftward shift of the force compensation curve compared with the learned force field curve for some of the directions. For test workspace 2 there is a leftward shift for some parts of the signal and a rightward shift for the rest of the signal compared with the learned force field. Similar to the joint-based representation, for object-based representation, the force compensation profiles change as the orientation of the hand is changed between workspaces. For both workspaces, the curves are shifted to the left with an increased leftward shift for test workspace 1. Thus, leftward shift of the force compensation curve for test workspace 1 compared to the learned forces curve would suggest that the representation does not rely on a Cartesian-based coordinate system. Similarly, leftward shift of the curve for test workspace 2 would suggest that the representation relies more on an object-based representation while a rightward shift would suggest that the representation relies more on a joint-based representation.

For experiment 2 (*Figure 4D and F*), the Cartesian based representation again predicts that the expected force profile in the test workspaces will be the same as the adapted force profile in the training workspace. In the case of a joint-based representation or an object-based representation, both predict a shift to the right with an increased rightward shift for test workspace 1.

For experiment 3 (*Figure 5D and F*), both the Cartesian-based and joint-based representations predict that the force compensation curves do not change or shift compared with the learned force field curve since the hand posture does not change between the training workspace and test workspaces. However, for the object-based representation, since the orientation of the pole is different between the training and test workspaces, the predictions of the force compensation curve are shifted to the right, more strongly so for test workspace 1. Thus, for this experiment, a rightward shift of the force compensation curves would suggest that representation relies more on an object-based representation.

For experiment 4 (*Figure 6D and F*), the Cartesian-based representation again predicts that the expected force profile in the test workspaces will be the same as the adapted force profile in the training workspace. Similarly, the object-based representation predicts the same expected force profile as hand orientation does not change between the training and test workspaces. However, since the arm configuration changes between these workspaces, the joint-based representation predicts alterations in the expected force profiles. For test workspace 1, joint-based representation predicts a shift to the left of the curve for some movement directions while for test workspace 2 the

prediction is a rightward shift of the curve. If instead there are small or no shifts of the curves then this would argue against a joint-based representation.

## Re-Dyn weight predictions

The reliability of each reference frame was defined as how much it is affected by noise and divert from a non-corrupted dynamic generalization pattern. To calculate the effect of additive neural noise on generated forces in the planning stage, we used the same calculations for each type of reference frame with added noise to the state variables. That is, for the Cartesian coordinate system, the noise was added to the end point coordinates $(x + \varepsilon_x, y + \varepsilon_y)$ and for the joint and object coordinate systems, we added noise to the shoulder and elbow joints values $(\theta_s + \varepsilon_s, \theta_e + \varepsilon_e)$, and by doing so, we also added noise to the hand orientation used by the object-based representation. There are multiple ways in which adding this noise can be implemented in a simulation. However, to have a comparable effect between coordinate systems, we used a simple scenario in which the noise that was added affected the start and target position of the movement. We added this noise to the movement at the test workspace according to each coordinate system. For the Cartesian coordinate system, the start coordinates $(x_s, y_s)$ were set according to $x_s = x_c + R \cdot cos(\alpha + \pi) + \varepsilon_{xs}$, $y_s = y_c + R \cdot sin(\alpha + \pi) + \varepsilon_{ys}$, and the target coordinates $(x_f, y_f)$ were set according to $x_f = x_c + R \cdot cos(\alpha) + \varepsilon_{xf}$, $y_f = y_c + R \cdot sin(\alpha) + \varepsilon_{yf}$.

For the joint and object coordinate system, the start coordinates $(x_s, y_s)$ were set according to $x_s = x_c + R \cdot cos(\alpha + \pi + \varepsilon_{xs})$, $y_s = y_c + R \cdot sin(\alpha + \pi + \varepsilon_{ys})$, and the target coordinates $(x_f, y_f)$ were set according to $x_f = x_c + R \cdot cos(\alpha + \varepsilon_{xf})$, $y_f = y_c + R \cdot sin(\alpha + \varepsilon_{yf})$.

The coordinates $(x_c, y_c)$ represent the center of the workspace and were set according to the value that was used in each experiment. $R$ represents the movement radius and α the movement direction. The random variables $(\varepsilon_{xs}, \varepsilon_{ys}, \varepsilon_{xf}, \varepsilon_{yf})$ represent the additive noise values to the start and target coordinates. For the joint and object coordinate systems the additive noise values were drawn from a normal distribution with zero mean and variance equal to 1°. This corresponds to ~1.7 mm variance in end-point coordinates (Cartesian coordinate system) for a movement radius of 10 cm. This calculation is based on an arc length formula $(Arc = R \cdot \theta)$ where θ is the central angle of the circle ($\theta$=1° ≈ 0.017 rad). Keeping this relation and setting it according to movement radius, we can ensure the noise level is comparable between all coordinate systems. Adding noise to these variables affected the different coordinate systems in multiple ways. In addition to changing the start and target position for all coordinate systems, for the joint-based representation it also changed the value of the Jacobian matrices and for the object-based system it changed the value of the rotation matrix. Since the Jacobian and rotation matrix also depend on movement direction, the noise has different effects in different directions. Keeping the same level of noise across all coordinate systems, regardless of the control variable, allowed us to examine to what extent noise affects the force profile generated by each representation considering the nonlinear nature of some of the coordinate systems.

For each test workspace and for each direction, we calculated the mean force value in two cases; with and without noise. We then calculated the error between these two values and repeated this process 1000 times for each direction to get the error distribution. We performed this process for the three reference frames, i.e., Cartesian, joint, and object. An example of this process is illustrated in *Figure 1*. From the error distributions, we extracted the variance for each and used the three variances, to calculate the inverse variance weights:

$$w_c = \frac{\frac{1}{\sigma_c^2}}{\frac{1}{\sigma_c^2} + \frac{1}{\sigma_j^2} + \frac{1}{\sigma_o^2}}; \quad w_j = \frac{\frac{1}{\sigma_j^2}}{\frac{1}{\sigma_c^2} + \frac{1}{\sigma_j^2} + \frac{1}{\sigma_o^2}}; \quad w_o = \frac{\frac{1}{\sigma_o^2}}{\frac{1}{\sigma_c^2} + \frac{1}{\sigma_j^2} + \frac{1}{\sigma_o^2}} \tag{5}$$

where $\sigma_c^2, \sigma_j^2$ and $\sigma_o^2$ are the error distribution variance for the Cartesian, joint, and object coordinate systems, respectively.

Using the predicted weights, we combined the force profile predicted by each reference frame. That is, according to the Re-Dyn model, the expected forces in the test workspaces are calculated as a weighted sum of the expected force values of the individual reference frames:

$$\boldsymbol{F} = w_c \cdot \boldsymbol{F}_{Cartesian} + w_j \cdot \boldsymbol{F}_{joint} + w_o \cdot \boldsymbol{F}_{object}. \tag{6}$$

Using this force profile, we calculated the force compensation in a similar way to the force compensation we calculated for the experimental results. Previous studies suggested that the magnitude of the generalized forces might decay as we move away from the training workspace (*Gandolfo et al., 1996*; *Berniker et al., 2014*). Thus, in order to match the observed behavioral force compensation patterns and the predicted force compensation pattern, we multiplied the force compensation profile of the combined model in (*Equation 6*) with a decay factor that set an equal amplitude between the predicted and experimental force compensation profiles:

$$\lambda = \frac{max\left(\boldsymbol{F}_{\text{exp}}\right) - min\left(\boldsymbol{F}_{\text{exp}}\right)}{max\left(\boldsymbol{F}_{\text{model}}\right) - min\left(\boldsymbol{F}_{\text{model}}\right)}.$$

(7)

where $\boldsymbol{F}_{\text{exp}}$ is the experimental force compensation profile and $\boldsymbol{F}_{\text{model}}$ is the predicted force compensation profile.

Similar to the phase shift analysis of the experimental force compensation curves, we calculated the phase shift of the predicted force compensation curve with decay. In addition, we calculated the root mean square error (RMSE) between the predicted and experimental force compensation curves across the 16 directions for all test workspaces.

## Energy-based weights predictions

A second model for predicting the weighting between reference frames we considered was the minimum energy criterion (*Nelson, 1983*; *Soechting et al., 1995*; *Berret et al., 2008*). According to this criterion, for each movement direction the motor system will rely more on reference frames which require less energy to produce the pattern of generalized forces. Since energy is proportional to the magnitude of required forces, we quantified the amount of energy needed by each reference frame as the force profile mean. That is, for each direction, we calculated the mean force value for a simulated generalized force profile for each reference frame, $Energy \sim \bar{F} = \frac{1}{N}\sum_{i=1}^{N}F\left[i\right]$, where F is the force profile according to the Cartesian, joints or object reference frames and N is the number of samples in the simulated trajectory. Similar to the Re-Dyn predictions, we used a normalized inverse energy quantity to calculate the weights, that is:

$$w_c = \frac{\frac{1}{|\bar{F}_c|}}{\frac{1}{|\bar{F}_c|} + \frac{1}{|\bar{F}_j|} + \frac{1}{|\bar{F}_o|}}; \quad w_j = \frac{\frac{1}{|\bar{F}_j|}}{\frac{1}{|\bar{F}_c|} + \frac{1}{|\bar{F}_j|} + \frac{1}{|\bar{F}_o|}}; \quad w_o = \frac{\frac{1}{|\bar{F}_o|}}{\frac{1}{|\bar{F}_c|} + \frac{1}{|\bar{F}_j|} + \frac{1}{|\bar{F}_o|}}$$

(8)

where $|\bar{F}_c|$, $|\bar{F}_j|$ and $|\bar{F}_o|$ are the absolute mean force value for the Cartesian, joint, and object reference frame, respectively. Similar to the Re-Dyn model, we used these weights to compute the force profile for each movement direction (according to *Equation 6*) which was used to compute the force compensation profile. The resulting force compensation profile was scaled by the decay factor followed by a computation of the phase shift and prediction RMSE.

## Smoothness-based weights predictions

The third model for predicting the weighting between reference frames was based on the maximum smoothness criterion. Smoothness was previously hypothesized to be an objective of the motor system when planning and executing movements (*Flash and Hogan, 1985*; *Berret et al., 2008*). Based on this hypothesis, we tested whether the motor system relies more on reference frames in which the force profile is smoother. For this purpose, we calculated the smoothness using the mean value of the third temporal derivative of the force profile predicted by each reference frame. That is, for each direction, we calculated the third derivative of the force signal and then calculated the mean value of it, $Smoothness = \bar{S} = \frac{1}{N}\sum_{i=1}^{N}\Delta^3 F\left[i\right]$, where F is the force profile according to the Cartesian, joint or object reference frames and N is the number of samples in the simulated trajectory. In this case, smoother movements have a lower mean value. The weight value is calculated according to the inverse mean value, that is

$$w_c = \frac{\frac{1}{|\bar{S}_c|}}{\frac{1}{|\bar{S}_c|} + \frac{1}{|\bar{S}_j|} + \frac{1}{|\bar{S}_o|}} \; ; \; w_j = \frac{\frac{1}{|\bar{S}_j|}}{\frac{1}{|\bar{S}_c|} + \frac{1}{|\bar{S}_j|} + \frac{1}{|\bar{S}_o|}} \; ; \; w_o = \frac{\frac{1}{|\bar{S}_o|}}{\frac{1}{|\bar{S}_c|} + \frac{1}{|\bar{S}_j|} + \frac{1}{|\bar{S}_o|}} \tag{9}$$

where $|\bar{S}_c|$, $|\bar{S}_j|$ and $|\bar{S}_o|$ are the mean absolute third derivative of force value for the Cartesian, joint, and object reference frame, respectively. We repeated the same phase and prediction error analysis procedures as we performed for the Re-Dyn and energy-based models.

### Fitted optimal weights model

The purpose of this model is to examine the extent of the three reference frames, i.e., Cartesian, joint, and object, to explain the experimental force compensation data. For this model, we searched for a single set of optimal weights, $w_c, w_j, w_o$, such that the squared error between the mean force profile of participants and the weighted predicted force, calculated using the simulations, is minimized across all movement directions. For this optimization problem, we constrained the weights to positive values and set the sum of the weights to be equal to or less than one. This constraint on the weights takes into consideration the decay of forces as we move away from the training workspace. After fitting the weights, we performed the phase and fitting RMSE analysis in a similar manner to the previous models.

### Bayesian information criterion (BIC)

Based on the error between the predicted and experimental force compensation profiles, we performed a BIC comparison between the models (Re-Dyn, energy, smoothness, and optimal). To calculate the BIC we used:

$$BIC = -2 \cdot log\left(L\right) + k \cdot log\left(n\right)$$

where log(L) is the logarithm of the model likelihood, k is the number of degrees of the model (one for Re-Dyn, energy, smoothness due to the decay factor and three for the optimal model due to the fitted weights), and n=128 is the total number of data points (four experiments X 2 workspaces X 16 directions). We calculated the BIC for each model and compared it the to BIC for a 'no generalization' model (that is, a model in which the force is zero in all movement directions). This allowed us to identify the best models that explain the mean generalized force profiles while accounting for different degrees of freedom between the models that predict the weights (Re-Dyn, smoothness, and energy) and the direction-independent weights optimal model (that is fitted to the data).

### Re-Dyn model predication for previous studies

To find the predicted weights of the Re-Dyn model for previous studies, we repeated the same simulations similar to the simulations we performed based on the experimental design reported here while changing the training and test workspace locations and the type of force field according to each study. Workspaces locations were set based on the joints angles (*Figure 8*) for a two-link arm with lengths 33 and 34 cm for the upper arm and forearm, respectively, that were used in *Shadmehr and Mussa-Ivaldi, 1994*. The authors of the previous studies implemented a viscous force field, i.e., $\begin{bmatrix} F_x \\ F_y \end{bmatrix} = B \cdot \begin{bmatrix} \dot{x} \\ \dot{y} \end{bmatrix}$, with a different B matrix (N·s/m). For the study by *Shadmehr and Moussavi, 2000* we used B=$\begin{bmatrix} -11 & -11 \\ -11 & 11 \end{bmatrix}$, for the study by *Malfait et al., 2005* we used B=$\begin{bmatrix} -20 & -20 \\ 0 & 0 \end{bmatrix}$, for the study by *Criscimagna-Hemminger et al., 2003* we used B=$\begin{bmatrix} 0 & 13 \\ -13 & 0 \end{bmatrix}$ and for the studies by *Shadmehr and Mussa-Ivaldi, 1994* and *Berniker et al., 2014* we used B=$\begin{bmatrix} -10.1 & -11.2 \\ -11.2 & 11.1 \end{bmatrix}$.

## Additional information

### Funding
No external funding was received for this work.

### Author contributions
Raz Leib, Conceptualization, Data curation, Software, Formal analysis, Validation, Investigation, Visualization, Methodology, Writing – original draft, Project administration, Writing – review and editing; David Franklin, Conceptualization, Supervision, Investigation, Methodology, Writing – original draft, Project administration, Writing – review and editing

### Author ORCIDs
Raz Leib ⓘ https://orcid.org/0000-0002-3940-2651
David Franklin ⓘ https://orcid.org/0000-0001-9530-0820

### Ethics
Participants signed a consent form (that included consent for participating in the study and consent for data processing and publication). The experimental protocol was approved by the Ethics Committee of the Medical Faculty of the Technical University of Munich (number 763/20 S-KH).

### Decision letter and Author response
Decision letter https://doi.org/10.7554/eLife.84349.sa1
Author response https://doi.org/10.7554/eLife.84349.sa2

## Additional files

### Supplementary files
MDAR checklist

### Data availability
The data generated and/or analysed during the current study are available in Open Science Framework.

The following dataset was generated:

| Author(s) | Year | Dataset title | Dataset URL | Database and Identifier |
| --- | --- | --- | --- | --- |
| Raz L, David WF | 2022 | Reference Frame | https://osf.io/6y2ug | Open Science Framework, 6y2ug |

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

# Appendix 1

## Comparison of mixture models composition

In this part, we show that the mixture model including Cartesian, joint, and object coordinate systems was the best model that fits the data across all four experiments. We based these results on an analysis of direction-independent fitted weights. That is, fitting one set of representations weights that can capture the force compensation profile. Overall, we considered 14 models; three individual models (Cartesian, joints, object) with or without decay (total 6 models), and eight mixture models covering all combinations of the individual models with and without decay. To compare the performance of the models in explaining the mean generalized force profiles, we used the Bayesian information criterion (BIC, see methods for more details):

$$BIC = -2 \cdot log\left(L\right) + k \cdot log\left(n\right)$$

with k set to 1 for individual models and 2 or 3 for mixture models, and n=16 is the total number of data points. We fitted the models separately for the results of each test workspace for each experiment and afterward calculated the BIC for each model. We again compared the BIC results to a no-generalization model.

BIC results of the fitted models for the two test workspaces across the four experiments are depicted in *Appendix 1—figure 1*. The mixed model of Cartesian, joint- and object-based representation was the only model that was selected as one of the best-fitting models across all workspaces for the four experiments. While for some experiments other models can explain the results, the mixture of Cartesian, joint, and object coordinate systems was indistinguishable from these other models.

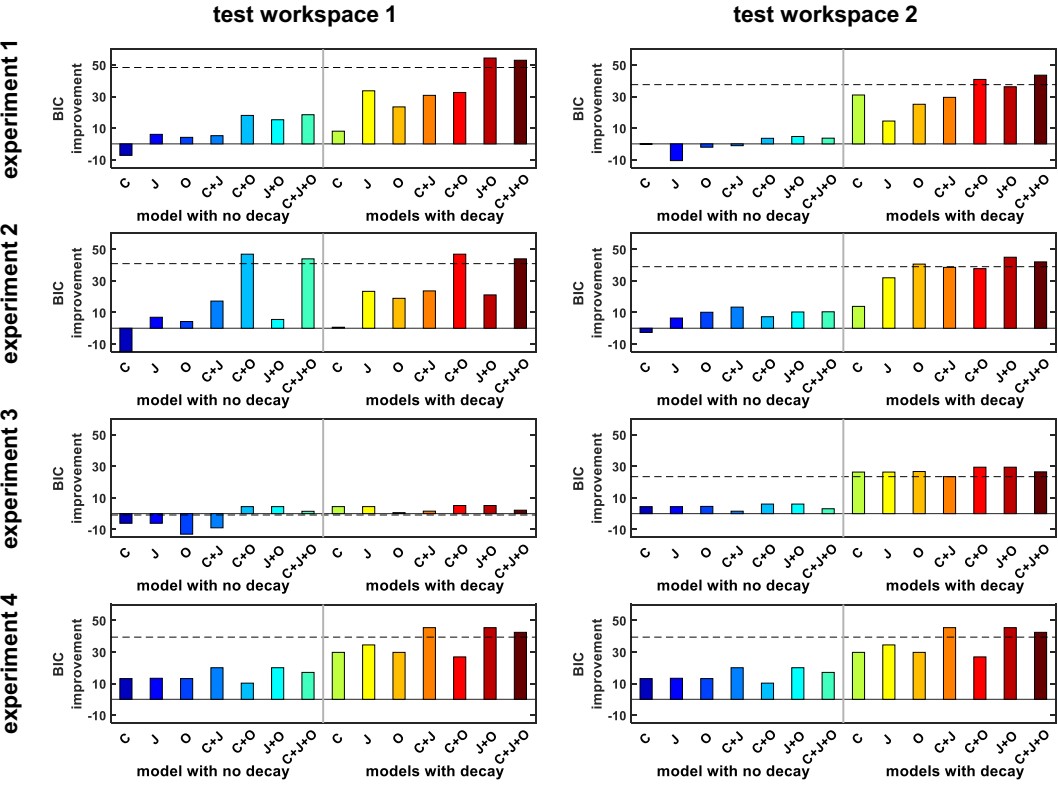

**Appendix 1—figure 1.** Bayesian Information Criterion (BIC) Improvement for each of the models relative to no generalization (that is, a model in which the force is zero in all movement directions). The dashed horizontal line shows the cutoff for models that are not considered distinguishable in terms of their performance from the best model.

## Appendix 2

### Testing ability of direction-independent weights to predict generalization errors

To better understand if a model based on direction-independent weights can explain the combination between coordinate systems, we perform an additional experiment. We used the idea of experiment 3 in which participants generalize learned dynamics using a tool. That is, the arm posture does not change between the training and test areas. In such a case, the Cartesian and joint coordinate systems do not predict a shift in generalized force pattern while the object coordinate system predicts a shift that depends on the orientation of the tool. In this additional experiment, we set a test workspace in which the orientation of the tool is 90° (*Appendix 2—figure 1A*). In this case, for the test workspace, the force compensation pattern of the object-based coordinate system is in anti-phase with the Cartesian/joint generalization pattern. Any direction-independent fitted weights (including equal weights) can produce either a non-shifted or 90° shifted force compensation pattern (*Appendix 2—figure 1B*). Participants in this experiment (n=7) showed similar MPE reduction as in all previous experiments when adapting to the trigonometric scaled force field (*Appendix 2— figure 1C*). When examining the generalized force compensation patterns, we observed a shift of the pattern in the test workspace of 14.6° (*Appendix 2—figure 1D*). This cannot be explained by the individual coordinate system force compensation patterns or any combination of them (which will always predict either a 0° or 90° shift, *Appendix 2—figure 1E*). However, calculating the prediction of the Re-Dyn model we found a predicted force compensation pattern with a shift of 6.4° (*Appendix 2—figure 1F*). The intermediate shift in the force compensation pattern suggests that any direction-independent weights cannot explain the results.

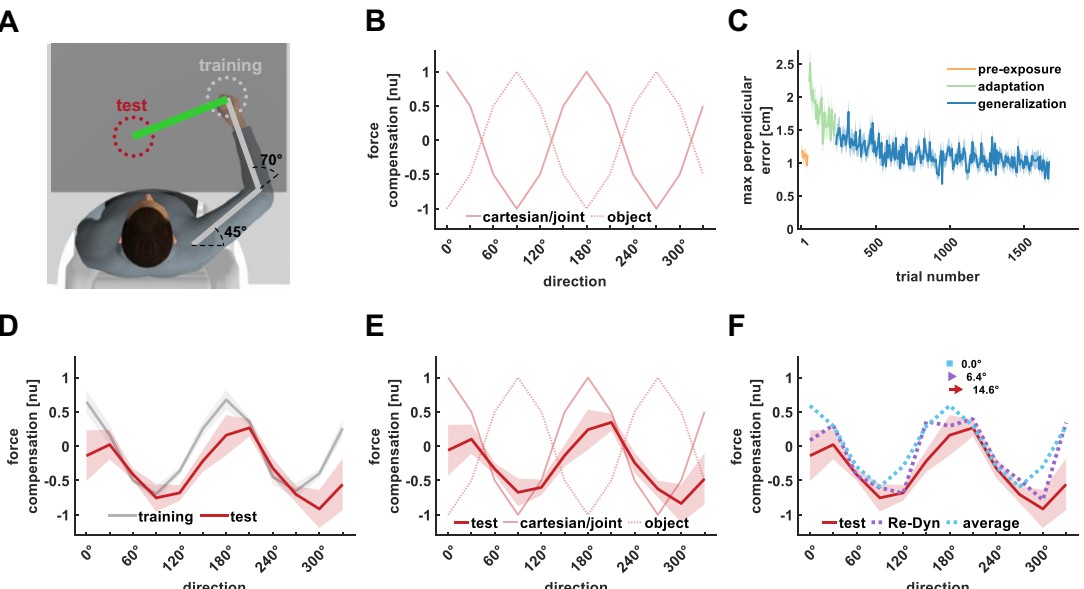

**Appendix 2—figure 1.** Direction-independent weights experiment. (**A**) Similar to experiment 3, the participants moved in the training workspace and the generalized forces were measured during movement of the endpoint of a 90° rotated pole in a test workspace. (**B**) Predicted generalized force compensation patterns for the Cartesian/ joint (solid line) and object-based (dotted line) coordinate systems for the test workspace.(**C**) Mean maximum perpendicular error (MPE) reduction during adaption to the trigonometric scaled force field. (**D**) Mean force compensation patterns and standard error (shaded area) in the training (gray) and test (red) workspaces. For the test workspace, the generalization pattern was shifted by 14.6°. (**E**) Mean force compensation profile for the test workspace (red solid line) and Cartesian/ joint- (light red solid line), and object- (light red dotted line) based model predictions. (**F**) Mean force compensation profile for the test workspace (red). To explain this pattern, we calculated the predicted force compensation profile generated by a mixture of reference frames according to the Re-Dyn predicted weights (purple dotted line) and a model based on direction-independent weights (light blue dotted line). In this case, the direction-independent weights model was calculated by setting equal weights between the three reference frames. Arrows at the top indicate the shift of the mean force compensation profile and the model's prediction.

