## [Editor Report]

This study provides a valuable new perspective on how motor learning occurring in one state generalizes to new states (for example, a different limb posture). The paper proposes a new model in which different potential coordinate systems for generalization are combined based on their relative reliability. The authors provide convincing evidence for this model, showing that it improves significantly on previous theories in its ability to predict patterns of generalization of motor learning in human subjects.

---

## [Decision Letter]

**Decision letter after peer review:**

Thank you for submitting your article "Error Prediction Determines the Coordinate System Used for Novel Dynamics Representation" for consideration by *eLife*. Your article has been reviewed by 3 peer reviewers, including Adrian M Haith as the Reviewing Editor and Reviewer #1, and the evaluation has been overseen by Tamar Makin as the Senior Editor.

The reviewers all agree that the paper provides a novel perspective on an outstanding problem in motor adaptation and that the direction-dependent weighting approach presents a promising way forward for advancing our understanding of generalization. The experiments are all well-designed and conducted and the empirical patterns of behavior in the results are very clear and the model predictions are well-aligned with the data in most cases. Overall, the reviewers felt that the paper has significant promise. However, they also identified some substantial concerns.

One major concern is that the model is quite weakly motivated and is also not described in sufficient detail. There were some concerns about the conceptual foundation of the model, including whether the analogy to cue combination is justified. A clearer and more concrete presentation, including derivation of the model from underlying principles, would significantly improve the paper. Many aspects of the model were not presented clearly enough (more equations would be helpful) and some aspects of the model were hardly described at all (e.g. the 'decay' factor), making it difficult to completely evaluate the model.

A second set of concerns relates to the strength of evidence for the model. In most experiments, the model predicts participants' behavior quite well. However, ultimately the current evidence is correlational and there is no specific, critical test of the noise-based weighting model or its underlying assumptions. The conclusion currently rests on a model comparison between the noise-based weighting model and two alternative models. However, the alternative models considered are far from exhaustive. It seems very possible that the data could be equally well described by other models in which the weighting between coordinate frames is direction dependent, but based on a different premise. Indeed, the 'optimal' fitted model performs significantly better than the noise-based model. In general, there were concerns about the extent to which the model could be falsified, and about whether this specific proposal about how direction-dependent weights are obtained could be disambiguated from potential alternatives.

While we consider these to be significant weaknesses of the paper, it does seem that they may potentially be addressable. We would, therefore, be willing to consider a revised version of the manuscript. In this revision, we would expect you to:

1) Provide a clearer rationale for and presentation of the model. This should include a clear statement of any underlying assumptions (e.g. potentially normative principles) and derivation of the weighting rule from these. Mathematical details should also be more concretely provided in the form of equations.

2) Provide stronger evidence that alternative theories could not account for observed patterns of generalization equally well or better. This could be in the form of additional analyses of the existing data, or new data that test a more critical prediction of the noise-based weighting model.

*Reviewer #1 (Recommendations for the authors):*

The authors claim, on line 320, that: "Importantly, we show that the weight for each coordinate system is inversely proportional to the force error produced when using a specific coordinate system." I don't think this was shown. It was the core assumption behind the model, rather than something which was shown empirically. Actually showing this would require estimating the weights for each coordinate system, which wouldn't actually be possible when there are three coordinate systems. This gives two free parameters and only 1 observation per direction.

The weights for each coordinate frame vary as a function of direction in a complex and non-intuitive way. It would be very helpful to provide some better insight as to how these patterns arise. For instance, as an accompaniment to Figure 1, it would be useful to see how the variance for each coordinate frame varies with direction, and then how this is converted into a weighting.

The presentation of the theory inconsistently describes the noise sometimes as motor execution noise, and sometimes as noise in the estimation of the initial state of the limb. Some parts of the paper suggest that the noise is added to control variables (e.g. line 631, line 340), while the more detailed description in the methods suggests it is added to state (e.g. 632). These two concepts aren't the same, but they are used interchangeably. The paper needs to be clearer about this.

line 324: "Our idea suggests that the motor system is minimizing the variability in extrapolated force production". It's not clear to me how the proposed method achieves minimization of variability.

It's not clear how the 'same level of noise' was achieved across all coordinate systems. What was the exact criterion for setting the noise values? I don't think it's obvious how to equate them across coordinate systems.

line 662 – "we multiplied the force compensation profile of the combined model in [6] with a decay factor that was equal to the amplitude of the experimental force compensation profile". I don't find this clearly enough explained. It needs to be written out mathematically. If there is a decay factor that increases with distance from the learning location, shouldn't this be specific to each coordinate system, rather than applied to the composite force output?

In many cases, the weights and model predictions seem to fluctuate quite substantially across directions – more rapidly than the 22.5 deg separation between probe directions. It seems that it should be possible to generate predictions in a more fine-grained manner than this, and this might be helpful in identifying conditions that would provide a more stringent test of the model, as well as helping to strengthen intuition about how and why these weights vary as they do.

It may be interesting to consider how the model would compare to a model with random weights, perhaps constrained to be periodic.

*Reviewer #2 (Recommendations for the authors):*

I think the figures would look better if almost all the top and rightmost axis lines were removed.

*Reviewer #3 (Recommendations for the authors):*

1) To make this paper significant, I recommend the author also write the derivation process for the proposed optimization principle.

2) I recommend the author elaborate the theory more to refute the possibility that the model may not be falsifiable [based on the concerns outlined in the public review].

3) I recommend the author write the details of these models so that we can understand these models better.

4) The detail of alternative models (energy and smoothness) are not written enough for us to understand how fair these comparisons are. Even the definition of absolute bar_F is not clear. Is this average across directions? Similar plots of the weighting factor over the angle, such as Figure *E is necessary for these alternative models.

5) Figure 5E Top. The weight of the joint coordinate is not plotted. Even in this case where the joint does not move, the w_j should be computed with Equation 5. I could not find any description of this omission of w_j. Similarly was not plotted in Figure 6.

6) The mathematical definition of the object-based coordinated system is not clear enough. According to L581-587, this angle is determined by the angle of the hand. However, this definition is different from the object-based coordinate, which can be manipulated by the visual object as defined in Ahmed 2008. Calling this hand angle the object angle is misleading. Also, the hand angle is determined by the wrist. The simple examination of this hypothesis is to change the wrist angle between the training space and the test spaces. Was the angle of the wrist fixed by a splint? If so, what is the purpose of constraining it? I recommend the authors describe this point.

[Editors' note: further revisions were suggested prior to acceptance, as described below.]

Thank you for resubmitting your work entitled "Error Prediction Determines the Coordinate System Used for the Representation of Novel Dynamics" for further consideration by *eLife*. Your revised article has been evaluated by Tamar Makin (Senior Editor) and a Reviewing Editor.

The model is explained much more clearly in the revised manuscript, and important details that were initially missing are now included. The majority of concerns raised by the reviewers have been addressed. The importance of direction-dependent weighting of coordinate frames is made very clear in the new experiment, and the Re-Dyn model represents a very reasonable and parsimonious version of a model that allows for this. One remaining concern relates to the rationale behind the model, which warrants a minor but important revision:

The revised manuscript now describes the model as related to "inverse optimal control", which I don't think is accurate and in fact it will be confusing to many readers. Inverse optimal control refers to the process of an observer trying to infer the cost function that is being optimized by an agent (as is the case in Berret et al. 2011). Lines 55-58 don't accurately describe the objective of inverse optimal control. The analogy to previous work on inverse optimal control seems more based on the specific approach in the Berret paper (estimating weights in a mixture model) rather than the actual problem being solved (determining mixture weights of a cost function in inverse optimal control, versus determining mixture weights for a policy/solution in this work).

So I don't believe the revision has entirely successfully addressed the need to "provide a clearer rationale for the model". The primary concern expressed previously was that there was no clear justification or derivation of the inverse-variance weightings – it was simply introduced in an ad-hoc manner, rather than being argued to stem a more fundamental principle such as minimizing a cost (note that this is the "derivation" the reviewers were asking for, not an analytical expression for the exact mixing weights). The crux of this issue is seen on line 77: "the motor system can set the relative contribution of each coordinate system by assigning different weights to each coordinate system, for example, using an inverse variance estimation" – and there is no further justification than this. It seems quite straightforward to justify the choice of inverse-variance estimation, but it is not explicitly stated anywhere and should be. For instance, one could simply argue that the motor system estimates/predicts the required forces based on three possible coordinate systems; there is noise in these predictions, and inverse-variance weighting is the optimal way to combine them in the sense of minimizing expected squared error or variance in the force output.

---

## [Author Response]

Essential revisions:The reviewers all agree that the paper provides a novel perspective on an outstanding problem in motor adaptation and that the direction-dependent weighting approach presents a promising way forward for advancing our understanding of generalization. The experiments are all well-designed and conducted and the empirical patterns of behavior in the results are very clear and the model predictions are well-aligned with the data in most cases. Overall, the reviewers felt that the paper has significant promise. However, they also identified some substantial concerns.

We would like to thank the editor and reviewers for their thorough review. In the rest of this file, we address the reviewers’ concerns with additional analysis and clarifications regarding the model. In addition, we took fully into consideration the reviewers’ suggestions and conducted additional experiments to address the issue with alternative models.

One major concern is that the model is quite weakly motivated and is also not described in sufficient detail. There were some concerns about the conceptual foundation of the model, including whether the analogy to cue combination is justified. A clearer and more concrete presentation, including derivation of the model from underlying principles, would significantly improve the paper. Many aspects of the model were not presented clearly enough (more equations would be helpful) and some aspects of the model were hardly described at all (e.g. the 'decay' factor), making it difficult to completely evaluate the model.

We do not consider this problem as a cue combination problem but rather as an inverse optimal control model. In such problems, different solutions which are based on different underlying principles are combined to explain a variety of behaviors. For example, in reaching movement to explain trajectory invariances such as the bell shape velocity profile, there were numerous feedforward based models, such as the minimum hand jerk or joint torque models, suggested to explain the optimal principle that forms the basis of these movements. While each model has its justification, it was suggested that the brain plans a movement by combining costs rather than using a single optimality criterion. In such a case, the inverse optimal control problem asks how different trajectories are combined, i.e., assuming a linear combination between models, and what weight is given to each model (Berret et al. 2011, Oguz et al. 2018). The studies trying to solve the inverse optimal control problem usually fitted the weights between models and tried to explain the fitted results. However, in our study, we suggest a prediction of the weights rather than fitting the weights. That is, given multiple ways to represent dynamics, we suggest an optimal way that predict the weighted sum between different dynamics representations.

Regarding the decay factor, we added an equation to describe it in addition to the explanation (see response to Reviewer 1). Importantly, the decay factor is an empirical result that was shown in the past and is not a feature of any particular model but a feature characterizing all the models to explain dynamics generalization. In our study, we tested a few models in addition to the minimum noise model, i.e. minimum energy, minimum smoothness, and fitted optimal model. For each of these models, we multiplied the predicted force compensation profile by the decay factor so it would match the experimental results in amplitude. In addition, the force profile we used in this study made the results robust to the decay factor since we compared the angular shift of the force profile using cross-correlation. In this case, neglecting the decay factor (or using any other calculation for it), will not affect the presented results since they do not depend on the signal’s amplitude.

A second set of concerns relates to the strength of evidence for the model. In most experiments, the model predicts participants' behavior quite well. However, ultimately the current evidence is correlational and there is no specific, critical test of the noise-based weighting model or its underlying assumptions. The conclusion currently rests on a model comparison between the noise-based weighting model and two alternative models. However, the alternative models considered are far from exhaustive. It seems very possible that the data could be equally well described by other models in which the weighting between coordinate frames is direction dependent, but based on a different premise. Indeed, the 'optimal' fitted model performs significantly better than the noise-based model. In general, there were concerns about the extent to which the model could be falsified, and about whether this specific proposal about how direction-dependent weights are obtained could be disambiguated from potential alternatives.

We conducted a new experiment in which we showed that a global optimal fitting of weights cannot explain the idea of dynamics generalization while using objects. This provides more support to a direction-based weight calculation. We then compared the three direction-based models (i.e., noise, energy, smoothness) to a random model that is based on random direction based weights and showed that while each of the meaningful models does better predicting the results, the noise-based model is the superior model out of these three. For the details, please see the response to Reviewer 1.

While we consider these to be significant weaknesses of the paper, it does seem that they may potentially be addressable. We would, therefore, be willing to consider a revised version of the manuscript. In this revision, we would expect you to:1) Provide a clearer rationale for and presentation of the model. This should include a clear statement of any underlying assumptions (e.g. potentially normative principles) and derivation of the weighting rule from these. Mathematical details should also be more concretely provided in the form of equations.

We thank the reviewers for bringing up this important aspect of the model. We now elaborated on the rationale for the model not as a multisensory combination problem but rather as an inverse optimal control problem with predicted rather than fitted weights (please see more details in the response to Reviewer 1). Due to the periodicity and nonlinearities of different force generalizations, an analytical derivation is not possible. We elaborated on this issue in the response to Reviewer 3. In addition, we added mathematical details regarding the numerical solution we used as an alternative to the analytical derivation.

2) Provide stronger evidence that alternative theories could not account for observed patterns of generalization equally well or better. This could be in the form of additional analyses of the existing data, or new data that test a more critical prediction of the noise-based weighting model.

We followed this suggestion and now report the results of another experiment in which we test the predictions of the average weighting between dynamics representations. The additional experiment, as reported in the answer to Reviewer 1, not only rejects the idea of average weights but also suggests that any model based on global weights (including fitted global weights) could not explain the new results. This supports our idea that weights are set locally according to the noise-based model.

Reviewer #1 (Recommendations for the authors):The authors claim, on line 320, that: "Importantly, we show that the weight for each coordinate system is inversely proportional to the force error produced when using a specific coordinate system." I don't think this was shown. It was the core assumption behind the model, rather than something which was shown empirically. Actually showing this would require estimating the weights for each coordinate system, which wouldn't actually be possible when there are three coordinate systems. This gives two free parameters and only 1 observation per direction.

We changed the phrasing of this sentence:

“Importantly, we show that the weight for each coordinate system was best predicted by a model that asserts that the weights are inversely proportional to the force error produced when using a specific coordinate system.”

The weights for each coordinate frame vary as a function of direction in a complex and non-intuitive way. It would be very helpful to provide some better insight as to how these patterns arise. For instance, as an accompaniment to Figure 1, it would be useful to see how the variance for each coordinate frame varies with direction, and then how this is converted into a weighting.

The reviewer is right with his observation that the weights pattern is complex and not intuitive. In addition to direction changes in the pattern, the pattern changes according to the arm configuration for the training and test areas. That is, a single pattern, even a detailed one, would describe only a specific scenario that depends on the set workspaces. The reason for that is the dependency of joint and object based coordinate systems on both the training and test workspaces. We added to Figure 1 an example of the variability across different directions for a specific case of the training and test workspaces.

The presentation of the theory inconsistently describes the noise sometimes as motor execution noise, and sometimes as noise in the estimation of the initial state of the limb. Some parts of the paper suggest that the noise is added to control variables (e.g. line 631, line 340), while the more detailed description in the methods suggests it is added to state (e.g. 632). These two concepts aren't the same, but they are used interchangeably. The paper needs to be clearer about this.

We thank the reviewer for pointing out this issue. We corrected the mentioned sentences and other places in the text. For example:

New line 349: “We used simulations of neural noise added to different state variables that define the controlled forces to estimate the error between desired and corrupted force output during generalization.”

New line 644: “To calculate the effect of additive neural noise on generated forces in the planning stage, we used the same calculations for each type of reference frame with added noise to the state variables.”

line 324: "Our idea suggests that the motor system is minimizing the variability in extrapolated force production". It's not clear to me how the proposed method achieves minimization of variability.

We corrected this sentence:

“Our idea suggests that the motor system uses a mix of force representation with a greater weight to the representation that is less variable in extrapolated force production.”

It's not clear how the 'same level of noise' was achieved across all coordinate systems. What was the exact criterion for setting the noise values? I don't think it's obvious how to equate them across coordinate systems.

To clarify this issue, we elaborate on the idea of adding noise to the simulation in the Methods section:

“The reliability of each reference frame was defined as how much it is affected by noise and divert from a non- corrupted dynamic generalization pattern. To calculate the effect of additive neural noise on generated forces in the planning stage, we used the same calculations for each type of reference frame with added noise to the state variables. That is, for the Cartesian coordinate system the noise was added to the end point coordinates (x+εx,y+εy) and for the joint and object coordinate systems, we added noise to the shoulder and elbow joints values (θs+εs,θe+εe) and by doing so, we also added noise to the hand orientation used by the object based representation. There are multiple ways in which adding this noise can be implemented in a simulation. However, to have comparable effect between coordinate systems, we used a simple scenario in which the noise that was added affected the start and target position of the movement. We added this noise to the movement at the test workspace according to each coordinate system. For the Cartesian coordinate system, the start coordinates (xs,ys) were set according to

xs=xc+R⋅cos⁡(α+π)+εxs,ys=yc+R⋅sin⁡(α+π)+εys

*,*

and the target coordinates (xf,yf) were set according to,

xf=xc+R⋅cos⁡(α)+εxf,yf=yc+R⋅sin⁡(α)+εyf. For the joint and object coordinate system, the start coordinates (xs,ys) were set according to xs=xc+R⋅cos⁡(α+π+εxs),ys=yc+R⋅sin⁡(α+π+εys), and the target coordiantes (xf,yf) were set according to xf=xc+R⋅cos⁡(α+εxf),yf=yc+R⋅sin⁡(α+εyf).

The coordinates (xc,yc) represent the center of the workspace and were set according to the value that was used in each experiment. *R* represents the movement radius and α the movement direction. The random variables (εxs,εys,εxf,εyf) represent the additive noise values to the start and target coordinates. For the joint and object coordinate systems the additive noise values were drawn from a normal distribution with zero mean and variance equal to 1°. This corresponds to ~1.7mm variance in end-point coordinates (Cartesian coordinate system) for a movement radius of 10cm. This calculation is based on an arc length formula (Arc=R⋅θ) where θ is the central angle of the circle (θ=1° ≈ 0.017rad). Keeping this relation and setting it according to movement radius, we can ensure the noise level is comparable between all coordinate systems. Adding noise to these variables affected the different coordinate systems in multiple ways. In addition to changing the start and target position for all coordinate systems, for the joint based representation it also changed the value of the Jacobian matrices and for the object based system it changed the value of the rotation matrix. Since the Jacobian and rotation matrix also depend on movement direction, the noise has different effects in different directions.”

line 662 – "we multiplied the force compensation profile of the combined model in [6] with a decay factor that was equal to the amplitude of the experimental force compensation profile". I don't find this clearly enough explained. It needs to be written out mathematically. If there is a decay factor that increases with distance from the learning location, shouldn't this be specific to each coordinate system, rather than applied to the composite force output?

We added an equation describing the decay factor (new equation 7 in the Methods section).

“…in order to match the observed behavioral force compensation patterns and the predicted force compensation pattern, we multiplied the force compensation profile of the combined model in [6] with a decay factor that set an equal amplitude between the predicted and experimental force compensation profiles:

λ=max(Fexp)−min(Fexp)max(Fmodel)−min(Fmodel) where ***F**_exp_* is the experimental force compensation profile and ***F**_model_* is the predicted force compensation profile.”

The reviewer raises an interesting question that can be answered in future studies targeting the characteristics of the decay in generalized forces and how it relates to the motor plan according to the different reference frames. Regardless of the decay factor nature, we treated it as a scaling factor to match the model prediction (which is always unscaled in magnitude) and the experimental force compensation amplitude. Importantly, the scaling factor is independent of the reference frames prediction (which amplitude is between -1 and 1) and we used the same extracted value between the tested weighting models. Critically, the decay cannot affect the angular shift of the force compensation curves, and thus, it does not affect the results of the shift analysis we performed.

In many cases, the weights and model predictions seem to fluctuate quite substantially across directions – more rapidly than the 22.5 deg separation between probe directions. It seems that it should be possible to generate predictions in a more fine-grained manner than this, and this might be helpful in identifying conditions that would provide a more stringent test of the model, as well as helping to strengthen intuition about how and why these weights vary as they do.

We run a detailed direction simulation to increase the resolution of the weights. For example, in the following figure we plotted the detailed weights simulation for experiment 1, test workspace 1. However, importantly the resolution of the weights does not help predict conditions in which the model can be tested. Instead, the more direct way to do so is to find workspaces in which the generalization patterns based on the different coordinate systems differ significantly. In other words, if the generalization patterns for each coordinate system are similar, the predicted generalization pattern will not change significantly even with great fluctuations of the weights.

**Author response image 1. sa2fig1:** Detailed simulation of weights for experiment 1, workspace 1.

It may be interesting to consider how the model would compare to a model with random weights, perhaps constrained to be periodic.

We run such a simulation and, as expected, random weights cannot accurately predict the generalization patterns. We used three sine waves shifted by 1 (so weights could not be negative) and with the same frequency as our trigonometric scaled force field. The change between these sine waves was the phase. For example, three random waves: wc=1+sin⁡(2⋅α+π2)wc=1+sin⁡(2⋅α)wc=1+cos⁡(2⋅α+π3) where α is the movement direction.

We then divided each sine wave by the sum of the three waves so that each weight contribution would not exceed 1. The signals above will then transform into:

**Author response image 2. sa2fig2:** Weights of a random weights model.

**Author response image 3. sa2fig3:** RMSE analysis comparing global fitted optimal model, three prediction based models, and a random weights model.

Reviewer #2 (Recommendations for the authors):I think the figures would look better if almost all the top and rightmost axis lines were removed.

We thank the reviewer for this suggestion. We removed the axis lines and reduced the number of ticks of the axis so it will look better.

Reviewer #3 (Recommendations for the authors):1) To make this paper significant, I recommend the author also write the derivation process for the proposed optimization principle.

As explained above, we hope the reviewer sees the problem with deriving the analytical expression of the model.

2) I recommend the author elaborate the theory more to refute the possibility that the model may not be falsifiable [based on the concerns outlined in the public review].

We hope our answer to the comment in the public review answers this suggestion.

3) I recommend the author write the details of these models so that we can understand these models better.

We thank the reviewer for this suggestion. We have extensively detailed our new model and the coordinate systems. In addition, we now elaborate our explanation regarding the other models, specifically the derivation process (as detailed below).

4) The detail of alternative models (energy and smoothness) are not written enough for us to understand how fair these comparisons are. Even the definition of absolute bar_F is not clear. Is this average across directions? Similar plots of the weighting factor over the angle, such as Figure *E is necessary for these alternative models.

We elaborated the explanation regarding the alternative models:

In the methods section (for the energy based model):

“Since energy is proportional to the magnitude of required forces, we quantified the amount of energy needed by each reference frame as the force profile mean. That is, for each direction, we calculated the mean force value for a simulated generalized force profile for each reference frame Energy∼F¯=∑i=1NF[i], where F is the force profile according to the Cartesian, joints or object reference frames and N is the number of samples in the simulated trajectory.”

(for the smoothness based model):

“…we calculated the smoothness using the mean value of the third temporal derivative of the force profile predicted by each reference frame. That is, for each direction, we calculated the third derivative of the force signal and then calculated the mean value of it,

Smoothness = S¯=1N∑i=1NΔ3F[i], where F is the force profile according to the Cartesian, joints or object reference frames and N is the number of samples in the simulated trajectory. “

We added here the plots of weights across the four experiments for the minimum energy and maximum smoothness models. The weights fluctuations as a function of movement direction is similar in nature to the fluctuation of the minimum noise model (for example the periodicity of the signals), however, it does not predict the experimental results as well as the noise-based model.

**Author response image 4. sa2fig4:** Weights according to minimum energy model.

**Author response image 5. sa2fig5:** Weights according to maximum smoothness model.

5) Figure 5E Top. The weight of the joint coordinate is not plotted. Even in this case where the joint does not move, the w_j should be computed with Equation 5. I could not find any description of this omission of w_j. Similarly was not plotted in Figure 6.

In Figure 5, which describe the results of experiment 3, the predicted force generalization of the joint coordinate system is equal to the predicted pattern of the Cartesian coordinate system. In such a case, since there is no change in posture between the training and test workspaces, the weight of the joint coordinate system is equal to the weight of the Cartesian coordinate system. We calculated the sum of the two coordinate system weights (w_c_+w_j_) and this sum weight was used for the weighted sum between the Cartesian/joint and object coordinate system (F=(w_c_+w_j_)F_c/j_+w_o_F_o_). Similarly, in Figure 6 the prediction and thus the weight of the object coordinate system is equal to the prediction and weight of the Cartesian coordinate system. We added a clarification for this in the caption of the figure:

“(E) Upper panel, the predicted weights for the mixture model for each movement direction according to the Re-Dyn model. In this case, since the Cartesian and joint based generalized forces are identical, their weight is equal and thus we used a sum of the two weights (Cartesian + joint).”

6) The mathematical definition of the object-based coordinated system is not clear enough. According to L581-587, this angle is determined by the angle of the hand. However, this definition is different from the object-based coordinate, which can be manipulated by the visual object as defined in Ahmed 2008. Calling this hand angle the object angle is misleading. Also, the hand angle is determined by the wrist. The simple examination of this hypothesis is to change the wrist angle between the training space and the test spaces. Was the angle of the wrist fixed by a splint? If so, what is the purpose of constraining it? I recommend the authors describe this point.

The reviewer is right with their observation regarding the difference between the two studies. While Ahmed 2008 is defining their coordinate system as ‘object based’, the mathematical definition they were using is describing a joint based coordinate system.

During the experiments, the wrist was not fixed since it was found to have a small effect in previous studies (e.g., Berniker et al. 2014). We added this point in the Method section:

“During the experiment, we assume participants maintain a fixed wrist angle and thus we did not fix their wrist joint similar to previous studies (Shadmehr and Mussa-Ivaldi 1994, Shadmehr and Moussavi 2000, Malfait, Shiller et al. 2002, Malfait, Gribble et al. 2005). The wrist deviation during force field and generalization experiments was shown to be small and thus have negligible effect (Berniker, Franklin et al. 2014).”

[Editors’ note: what follows is the authors’ response to the second round of review.]

The model is explained much more clearly in the revised manuscript, and important details that were initially missing are now included. The majority of concerns raised by the reviewers have been addressed. The importance of direction-dependent weighting of coordinate frames is made very clear in the new experiment, and the Re-Dyn model represents a very reasonable and parsimonious version of a model that allows for this. One remaining concern relates to the rationale behind the model, which warrants a minor but important revision:The revised manuscript now describes the model as related to "inverse optimal control", which I don't think is accurate and in fact it will be confusing to many readers. Inverse optimal control refers to the process of an observer trying to infer the cost function that is being optimized by an agent (as is the case in Berret et al. 2011). Lines 55-58 don't accurately describe the objective of inverse optimal control. The analogy to previous work on inverse optimal control seems more based on the specific approach in the Berret paper (estimating weights in a mixture model) rather than the actual problem being solved (determining mixture weights of a cost function in inverse optimal control, versus determining mixture weights for a policy/solution in this work).

Thank you for this comment. We understand how our phrasing might be confusing. We now clarified our meaning.

We hypothesize that this inconsistency in results can be explained using a framework in which the motor system assigns different weights to each solution and calculates a weighted sum of these solutions. Usually, to support such a framework, previous studies found the weights by fitting the weighed sum solution to behavioral data (Berniker, Franklin et al. 2014). While we treat the problem similarly, we propose the Reliable Dynamics Representation (Re-Dyn) mechanism that determines the weights instead of fitting them.

So I don't believe the revision has entirely successfully addressed the need to "provide a clearer rationale for the model". The primary concern expressed previously was that there was no clear justification or derivation of the inverse-variance weightings – it was simply introduced in an ad-hoc manner, rather than being argued to stem a more fundamental principle such as minimizing a cost (note that this is the "derivation" the reviewers were asking for, not an analytical expression for the exact mixing weights). The crux of this issue is seen on line 77: "the motor system can set the relative contribution of each coordinate system by assigning different weights to each coordinate system, for example, using an inverse variance estimation" – and there is no further justification than this. It seems quite straightforward to justify the choice of inverse-variance estimation, but it is not explicitly stated anywhere and should be. For instance, one could simply argue that the motor system estimates/predicts the required forces based on three possible coordinate systems; there is noise in these predictions, and inverse-variance weighting is the optimal way to combine them in the sense of minimizing expected squared error or variance in the force output.

We followed this suggestion and added the justification for using the inverse variance estimation.

We propose that based on these distributions, specifically the variances of each distribution, the motor system can set the relative contribution of each coordinate system. One way of doing so is by assigning different weights to each coordinate system, for example, using an inverse variance estimation, which was shown to be an optimal way of combining different estimations to minimize the variance of the combined estimation (Shadmehr and Mussa-Ivaldi 2012).